# Drug-Drug Interactions of Direct Oral Anticoagulants (DOACs): From Pharmacological to Clinical Practice

**DOI:** 10.3390/pharmaceutics14061120

**Published:** 2022-05-24

**Authors:** Nicola Ferri, Elisa Colombo, Marco Tenconi, Ludovico Baldessin, Alberto Corsini

**Affiliations:** 1Department of Medicine, University of Padova, 35100 Padua, Italy; 2Department of Pharmacological and Biomolecular Sciences, University of Milan, 20133 Milan, Italy; elisa.colombo1@unimi.it (E.C.); alberto.corsini@unimi.it (A.C.); 3EDRA S.p.A., 20141 Milan, Italy; m.tenconi@lswr.it (M.T.); l.baldessin@lswr.it (L.B.)

**Keywords:** apixaban, dabigatran, rivaroxaban, edoxaban, drug-drug interaction, pharmacokinetics, pharmacodynamics

## Abstract

The direct oral anticoagulants (DOACs), dabigatran, rivaroxaban, apixaban, and edoxaban, are becoming the most commonly prescribed drugs for preventing ischemic stroke in patients with non-valvular atrial fibrillation (NVAF) and for the treatment and prevention of venous thromboembolism (VTE). Rivaroxaban was also recently approved for the treatment of patients with a recent acute coronary syndrome (ACS). Their use demonstrated to have a favorable risk-benefit profile, with significant reductions in stroke, intracranial hemorrhage, and mortality compared to warfarin, but with increased gastrointestinal bleeding. Nevertheless, their safety profile is compromised in multimorbidity patients requiring contemporary administration of several drugs. Comorbidity and polypharmacy have a high prevalence in elderly patients, who are also more susceptible to bleeding events. The combination of multiple treatments can cause relevant drug–drug interactions (DDIs) by affecting the exposure or the pharmacological activities of DOACs. Although important differences of the pharmacokinetic (PK) properties can be observed between DOACs, all of them are substrate of P-glycoprotein (P-gp) and thus may interact with strong inducers or inhibitors of this drug transporter. On the contrary, rivaroxaban and, to a lower extent, apixaban, are also susceptible to drugs altering the cytochrome P450 isoenzyme (CYP) activities. In the present review, we summarize the potential DDI of DOACs with several classes of drugs that have been reported or have characteristics that may predict clinically significant DDIs when administered together with DOACs. Possible strategies, including dosage reduction, avoiding concomitant administration, or different time of treatment, will be also discussed to reduce the incidence of DDI with DOACs. Considering the available data from specific clinical trials or registries analysis, the use of DOACs is associated with fewer clinically relevant DDIs than warfarin, and their use represents an acceptable clinical choice. Nevertheless, DDIs can be significant in certain patient conditions so a careful evaluation should be made before prescribing a specific DOAC.

## 1. Introduction

Atrial fibrillation (AF) is the most common type of cardiac arrhythmia worldwide [1] and its prevalence is increasing due to the ageing population and other risk factors and comorbidities [2]. In the elderly, comorbidity and polypharmacy are quite common, and this population of patients showed a 9–10% incidence of AF at aged > 80 years, compared to less than 0.1% in patients at aged < 55 years) [3]. In AF patients, polypharmacy and multiple comorbidities have been associated to higher incidence of death and bleeding risk [4,5,6]. Moreover, the risk of drug–drug interactions (DDI) increases with the number of concomitant drug treatments [7]. 

Direct oral anticoagulants (DOACs) are considered by international guidelines as the preferred choice of anticoagulants to prevent stroke in patients with AF [2,8] and to prevent venous thromboembolism (VTE) [9]. Considering their large use, it is essential to have a complete picture on possible DDI between DOACs and other commonly used classes of drugs. Although the last European Heart Rhythm Association (EHRA) guidelines have nicely summarized the main DDI with DOACs [2], in the present review we have extended and updated the current knowledge on this topic that might be useful for clinicians for their prescriptions.

## 2. Pharmacokinetic and Pharmacodynamic Properties of DOACs

DOACs comprise dabigatran, a selective inhibitor of factor IIa (thrombin), and three factor Xa inhibitors: rivaroxaban, apixaban and edoxaban. Apixaban is the most potent inhibitor of factor Xa with an inhibitory constant (Ki) of 0.08 nM, 10,000 higher than thrombin (Ki ~ 3 μM) [10,11,12] compared with a Ki of 0.4 nM for rivaroxaban [13] and of 0.56 nM for edoxaban [14]. Differently, dabigatran acts at a lower level of the coagulative cascade by inhibiting the activity of thrombin with a Ki of 4.5 nM [10]. 

Considering their direct inhibitory action of factor Xa or IIa, the DOACs anticoagulant effect is linear with the plasma concentration of the drugs, with a maximal effect reached after approximately 3 h post oral administration (Tmax, Table 1). Results of clinical studies clearly show that inhibitors of factor Xa, rivaroxaban, apixaban, and edoxaban exert their pharmacological effect in a concentration-dependent manner [15,16,17,18,19,20,21]. The half-life of DOACs (5 ÷ 17 h) allows the reactivation of the coagulation cascade 12–24 h after the interruption of the therapy [21,22]. These pharmacological characteristics yield therapy with DOACs less problematic compared to warfarin, which has a very long onset of action (days), long half-life (20 ÷ 60 h), and the coagulation requires many days to be restored after interruption of therapy [23]. It is also relevant to point out that apixaban, edoxaban, and rivaroxaban are administered in active form, while dabigatran as a prodrug in the etexilate form to improve its bioavailability. 

Beyond the pharmacodynamic (PD) characteristics, the pharmacokinetic (PK) profile represents a second level of differentiation between DOACs and warfarin (Table 1). Considering the different parameters, those that more likely influence the interaction with other drugs are the interaction with P-glycoprotein (P-gp), cytochrome P450 (CYP450)-mediated metabolism, variability of plasma concentration of the drug (peak/through ratio), and renal elimination. From all these considerations, it is evident that DOACs have intrinsic characteristics which are deeply distinct, and thus their propensity to undergo to DDI may vary between specific drugs.

The PK parameters of DOACs are summarized in Table 1. Protein binding are remarkably different between DOACs, with rivaroxaban and apixaban showing values above 90% and 87%, respectively. Thus, these two drugs may undergo to protein displacement by drugs with higher affinity to albumin and possible increase of their exposure. We recall that warfarin shows a very similar high protein binding (89%), and *S*(−) isomer has a slightly greater affinity than *R*(+) [24]. P-gp plays an important role in PK profile of all DOACs by impairing their intestinal absorption and promoting elimination by the kidney and the liver (Table 1) [25,26]. For this reason, potent P-gp inhibitors or inducers (Table 2) are expected to have relevant pharmacological interactions with all DOACs, increasing or reducing their anticoagulant effect. 

Dabigatran differs from the other DOACs for its low bioavailability (6.5%) which determines a large variability in the quote of the drug absorbed at gastrointestinal (GI) level [11]. 

The presence of food slightly delays dabigatran absorption (C_max_ from 2 h to 4 h), while a significantly increase of rivaroxaban bioavailability is observed, especially at 20 mg dose [27]. In addition, the bioavailability of rivaroxaban is not linear with the administered dose, it is estimated to be 80–100% at the dose of 10 mg and 66% at 20 mg [27]. Differently, the GI absorption of apixaban and edoxaban is not influenced by the presence of food [17,18,28,29].

**Table 2 pharmaceutics-14-01120-t002:** Inducers and inhibitors of CYP3A and P-gp. Modified from Corsini et al. [30].

	P-gp Inhibitor	Non-P-gp Inhibitor	P-gp Inducer
**Strong CYP3A inhibitor**	itraconazole, ketoconazole, clarithromycin, lopinavir, indinavir, ritonavir, telaprevir	voriconazole	
**Moderate CYP3A inhibitor**	erythromycin, verapamil, diltiazem, dronedarone	not identified	doxorubicin
**Weak CYP3A inhibitor**	lapatinib, quinidine, cyclosporine, felodipine, azithromycin, ranazoline, ticagrelor, chloroquine, hydroxychloroquine	cimetidine	vinblastine
**CYP3A Inducers**			carbamazepine, phenytoin, phenobarbital, rifampin, dexamethasone, tocilizumab, St. John’s Wort

CYP = Cytochrome P 450; P-gp = P-glycoprotein.

Both apixaban and rivaroxaban are partially metabolized in the liver by CYP3A4; therefore, a strong inducer and inhibitors of this hepatic cytochrome (Table 2) may influence the PK of the anticoagulants and alter their pharmacological effect. 

Rivaroxaban, due to the once daily posology, and dabigatran, due to the low bioavailability, are expected to have a higher variability of plasma concentrations (Table 1) and may undergo more easily to clinically relevant DDI.

Anti-fXa chromogenic assays are available to measure plasma concentrations of DOACs. This determination may help clinicians to detect a DDIs [2]. 

In addition to the PK interactions, different classes of drugs may have a PD interaction by affecting either the coagulation cascade or platelet activation. Thus, the DDI can be divided in PK and PD according to the different mechanism of interaction.

The aim of this review was to summarize the known or predicted PK and PD interactions of DOACs with different classes of drugs, discussing the clinical relevance and possible strategies to reduce the incidence of these interactions.

## 3. Potential Drug–Drug Interaction with Antiarrhythmic Drugs

AF patients are commonly treated with cardiovascular drugs that might interact with DOAC through the inhibition of P-gp and/or CYP3A4, thus leading to increased exposure and bleeding risk [2]. 

Digoxin is a cardiac glycoside used for the treatment of congestive heart failure. Only 16% of digoxin is metabolized, while 50–70% is eliminated unmodified with urine. Digoxin has a narrow therapeutic index, and it is a substrate of P-gp. A clinical PK study clearly demonstrated that digoxin does not interact with apixaban (Table 3) [31]. Similarly, no significant changes of the PK profile of dabigatran are observed with digoxin, and thus a negligible impact was observed on blood coagulation time, activated partial thromboplastin time (aPTT), and ecarin clotting times (ECT) [32]. The same results were obtained in healthy subjects after co-administration of digoxin and rivaroxaban [33] with only a minor effect that has been observed in one study on the PK of edoxaban [34]. 

Atenolol is equally eliminated unmodified through the feces and urine and it is not an inhibitor of CYP450 and P-gp. Beta-blockers may interact with other drugs by reducing liver flow, although this should not affect the clearance of apixaban since it has a low hepatic extraction. This hypothesis was confirmed in a PK study [31]. The interaction between atenolol and other DOACS was not investigated clinically; however, atenolol is not expected to alter the PK of other DOACs.

Dronedarone, an antiarrhythmic drug with properties of Vaughn–Williams classes I-IV, is a strong inhibitor of P-glycoprotein and a moderate inhibitor of CYP3A4 [35]. As expected, dronedarone increases C_max_ and AUC of dabigatran etexilate 150 mg bid by 1.73-fold and 2-fold, respectively [36]. Dronedarone doubles dabigatran AUC and C_max_, thus its co-administration is contraindicated [36], as confirmed by a retrospective cohort study that observed a modest increased risk of GI bleeding but not overall bleeding in AF patients treated with dronedarone and dabigatran [35]. In the US, patients with values of clearance of creatinine between 30 and 50 mL/min, the combination dabigatran dronedarone is permitted only at the lowest dose of 75 mg bid, further supporting the relevance of this interaction. 

Caution must be taken with amiodarone, quinidine, and verapamil, moderate and mild P-gp inhibitors. In healthy volunteers, amiodarone was shown to increase dabigatran bioavailability by approximately 50–60% [37]. This DDI can be considered clinically relevant considering also the long half-life time of amiodarone [38]. The effect of amiodarone on dabigatran exposure (AUC) seems less evident, with an AUC increase by 12% in AF patients [39]. In any case, significantly higher incidence of major bleeding has been reported after a co-administration of amiodarone in dabigatran treated patients compared to dabigatran alone [40].

Similarly to amiodarone, quinidine, another P-gp inhibitor, increases the bioavailability of dabigatran, both AUC and C_max_, by more than 50% [2]. The interaction between dabigatran and verapamil depends on the time of administration and the formulation of verapamil. Dabigatran exposure increases significantly when given within 2 h with an immediate-release formulation of verapamil (AUC and C_max_ are 143% and 179%, respectively, compared to dabigatran alone) [41]. On the contrary, a minor interaction was observed when dabigatran was given 2 h before a double dose of extended-release verapamil (AUC and C_max_ < 20% of increase) [41]. Since the half-life time did not change, the interaction was most likely related to the absorption of dabigatran, further supporting the role of intestinal P-gp on DDIs. 

In healthy subjects, the co-administration of digoxin did not affect rivaroxaban PK and PD [33]. Indeed, digoxin is not expected to interact with DOACs [2]. Rivaroxaban does not induce or inhibit any major CYP isoforms, including CYP3A4, or P-gp/Bcrp transporters. Dronedarone seems to have a moderate effect on rivaroxaban PK and the last EHRA guidelines indicated to avoid their concomitant prescription [2]. This conclusion has been reached after the results of a specific retrospective cohort study in patients with AF ≥ 18 years treated with DOACs [35]. The results of this study showed a significant increased risk of overall bleeding, GI bleeding, in patient treated with dronedarone and rivaroxaban [35].

Mendell J et al., reported the results from six PK studies evaluating the potential interactions between edoxaban and cardiovascular drugs [34]. The relevance of the interaction strongly depends on the degrees of P-gp inhibition. For instance, verapamil, quinidine, dronedarone, and amiodarone are potent P-gp inhibitors [42], increased the AUC of edoxaban by about 50–85% [34]. A clear contribution of P-gp inhibition on DDI with DOACs has been demonstrated by comparing the effect of intravenous and oral administration of quinidine on edoxaban exposure, 35% vs. 77% increase, respectively [34,43]. However, the interaction with verapamil and quinidine has not been considered clinically relevant and no dose reduction is required, although caution should be considered in the presence of other factors that might increase edoxaban exposure [2]. This final statement has been reached after the analysis of phase III clinical data [2]. The determination of plasma concentration of edoxaban, in a subgroup of patients of ENGAGE AF-TIMI 48 trial, demonstrated a significant interaction with amiodarone [44]. Specifically, the concentrations were 58.5 ± 53.2 ng/mL with amiodarone vs. 43.2 ± 41.1 ng/mL without amiodarone [44]. The last EHRA guidelines does not suggest reducing edoxaban dosage with the concomitant use of amiodarone [2] (Table 3). 

### Summary

Considering all antiarrhythmic drugs, the most critical appears to be dronedarone, with important differences between DOACs. The contraindication for rivaroxaban and dabigatran results from higher bleeding risk (rivaroxaban) and higher bioavailability (dabigatran) that may lead to a more relevant DDI. For verapamil, a clear and important interaction has been described for dabigatran, while the anti-fXa factors can be considered as co-treatment. 

## 4. Potential Drug–Drug Interaction with Antiplatelet and Antithrombotic Drugs

Coronary heart disease (CAD) is a common comorbidity in patients with AF, with an incidence of approximately 25–35% [45,46]. This high incidence is largely due to the multiple risk factors shared by these pathological conditions (e.g., obesity, hypertension, diabetes mellitus). It is estimated that approximately 10% of patients with recent percutaneous coronary intervention (PCI) have concomitant AF [47]. In addition, rivaroxaban is approved for the treatment of patients with recent ACS in combination with antiplatelet drugs [48]. For this reason, the interaction of DOACs with antiplatelet drugs are of clinical importance. In patients with AF undergoing PCI, it is recommended a triple therapy with aspirin, P2Y12 antagonist, and oral anticoagulation [49]. However, this therapy is associated with 3- to 4-fold increased risk of bleeding complications [50,51,52]. For the PD prospective analysis, the results of four dedicated RCTs have investigated the efficacy and safety of DOAC or warfarin with anti P2Y12 inhibitors in patients with AF and ACS undergoing PCI [51,53,54,55]. These trials showed that dual therapy with DOAC plus P2Y12 inhibitors (mainly clopidogrel) performs better in terms of risk of bleeding compared triple therapy with warfarin, aspirin, and a P2Y12 inhibitor. The bleeding risk reduction was mainly driven by receiving DOAC instead of warfarin a well as by omitting aspirin [51]. Considering the possible PK interaction with aspirin, 100 mg once a day for 5 days does not influence the PK of edoxaban [36], whereas higher dose of aspirin (325 mg) increased edoxaban AUC by 30% and C_max_ by 34% (Table 4) [56]. Although, the reason for this DDI is still unknown, a clinically relevant 2-fold increase in bleeding time has been observed with aspirin 100 mg (low dose), or aspirin 325 mg (high dose) in combination with edoxaban [56]. This effect is more likely due to a PD interaction between the two drugs [56]. 

Although rivaroxaban does not increase the antiplatelet effect of aspirin, and aspirin does not alter the effect of rivaroxaban on the inhibition of fXa activity [57], their combination in patient with venous thromboembolism (VTE), was associated to 1.5-fold higher incidence of major bleeding events [58]. 

Antiplatelet drugs are inhibitors (ticagrelor, naproxen) or substrates (clopidogrel, enoxaparin) of P-gp [59,60,61] (Table 4). Edoxaban, with or without concomitant use of clopidogrel and ticagrelor, showed similar relative efficacy and reduced bleeding compared to warfarin [62]. Nevertheless, a significant increase of bleeding risk due to a PD interaction is expected with all DOACs when administered together with antiplatelet drugs, as observed with dabigatran [55]. No PK interaction was observed when clopidogrel (75 mg once daily) and dabigatran (150 mg twice daily) where administered in healthy volunteers [63]. However, a single loading dose (300 mg or 600 mg) of clopidogrel increased dabigatran AUC and C_max_ by 30–40% [63]. 

The P-gp inhibitor, ticagrelor, increases the exposure of dabigatran by almost 50% (AUC and C_max_ +48.3% and +62.7%, respectively) (Table 4) [64]. A less evident interaction between the two drugs was observed when ticagrelor was administered 2 h after morning dose of dabigatran (AUC and C_max_ +28.8% and +24.1%, respectively). This staggered intake is clearly indicated in the SmPC [64]. A similar behavior can be predicted for the other DOACs. Finally, apixaban does not further inhibit platelet aggregation when administered with prasugrel (60 mg followed by 10 mg once daily) (Table 4) [50]. 

### Summary

All DOACs show an important and clinically relevant PD DDI with antiplatelet drugs. Ticagrelor is the only one that partially inhibit P-gp with a significant increase of drug exposure for dabigatran and potentially for the other DOACs. 

## 5. Potential Drug–Drug Interaction with Nonsteroidal Anti-Inflammatory Drugs

Long-term treatment with nonsteroidal anti-inflammatory drugs (NSAIDs) could be expected in patients with AF as they tend to be elderly and to have other inflammatory disorders. It is logical to predict a PD interaction between NSAIDs and DOAC with a significant increase in bleeding risk; indeed, their chronic use is not permitted by the respective SmPCs. This interaction has been documented in the EINSTEIN trial where rivaroxaban was compared to enoxaparin-vitamin K antagonists (VKA) treatment. The incidence of major bleeding during NSAID-anticoagulant treatment was equal to 6.5 per 100 patient-years, compared to 2.0 per 100 patient-years during anticoagulant use only (HR, 2.37) [58]. A similar increase was observed for clinically relevant bleeding (HR, 1.77) (Table 5) [58,65]. 

In a post hoc analysis of the RE-LY study, the use of NSAIDs was associated with increased risk of major bleeding (HR, 1.68) and GI bleeding (HR, 1.81), stroke/SE (HR, 1.50), and hospitalization (HR, 1.64) [66]. The safety and efficacy of dabigatran 150 and 110 mg b.i.d. relative to warfarin were not altered [66]. 

A PK study has found that the use of the nonselective NSAID naproxen increased serum concentrations of apixaban and could potentially increase the risk of bleeding (Table 5) [67]. A post hoc analysis of the ARISTOTLE trial that the incident NSAIDs use was associated with increased risk of clinically relevant nonmajor bleeding (HR, 1.70), major bleeding (HR, 1.61), but not GI bleeding [65]. However, NSAIDs use in patients with AF treated with apixaban relative to warfarin was not independently associated with an increased risk of bleeding or adverse events [65]. 

### Summary

Similarly to antiplatelet drugs, NSAIDs interact with DOACs by modulating the platelet activity. This interaction has been clearly observed in clinical trials. Naproxene has an additional mechanism of interaction by a specific competition on P-gp that has been detected with apixaban but can be considered for all DOACs. 

## 6. Potential Drug–Drug Interaction with Antidepressant Drugs

Antidepressants are widely used in the treatment of patients with stroke [68]. A retrospective cohort study conducted in patients with AF documented an increased risk of intracerebral hemorrhage in patients treated with the combination of DOACs with SSRIs (+38%), particularly with paroxetine and tetracyclic antidepressants [69]. These results, although from a retrospective study, indicate a clinically relevant DDI between DOACs and antidepressants, which should be carefully considered when prescribing DOACs in adult patients (Table 6).

St. John’s wort is one of the most commonly used remedies for minor and major depression [70]. The effect of St. John’s wort is a strong induction of P-gp and CYP3A4 and can potentially affect the anticoagulant action of all DOACs [71]. Due to the frequent use of this substance, the non-standardized dosages and the expected reduction of plasma concentrations with all DOACs, its use should be avoided in concomitance with DOACs (Table 6) [72,73]. In an open-label, nonrandomized, sequential treatment interaction study conducted with 12 healthy volunteers, St. John’s wort extract significantly reduced the AUC and Cmax of rivaroxaban by 24% and 14%, respectively. No clinically significant differences were found regarding T_max_ and half-life of rivaroxaban. Thus, St. John’s wort extract significantly interact with rivaroxaban and are predicted to interfere with other DOACs by inducing CYP3A4 and P-gp expression [74].

### Summary

The most relevant interaction relies on St. John’s wort, a strong inducer of P-gp and thus determining a clinically significant reduction of DOACs activity. The effect of SSRI and clomipramine are less clear as their inhibitory effect on platelet aggregation is still controversial. 

## 7. Potential Drug–Drug Interaction with Statins and Lipid-Modified Agents

Lipid-modifying agents are widely utilized in AF patients, in consideration to the high rate of coronary heart disease (CHD). Statins have some effect on P-gp and CYP450 [36]. Atorvastatin, lovastatin, and simvastatin are metabolized by CYP3A4 and may compete with P-gp [75] and thus might increase the exposure of DOACs. This effect can be considered clinically relevant in consideration to the results of a population-based, nested case-control study involving 45,991 Ontario residents under treatment with dabigatran. A higher risk of major hemorrhage was observed in patients under treatment with dabigatran and simvastatin or lovastatin compared to dabigatran alone (OR 1.46) [76]. Similar effect can be predicted for the other DOACs. Opposite results were observed with atorvastatin in the analysis of the Taiwan National Health Insurance database. In patients cotreated with atorvastatin and dabigatran, the adjusted incidence rate for major bleeding was significantly lower than dabigatran alone [40]. Specific PK study clearly demonstrated a lack of interaction between dabigatran and atorvastatin [77]. These results were confirmed by a second PK study were no significant differences was observed in the C_trough_ and C_max_ concentration of dabigatran in the presence or absence of atorvastatin [78]. Similarly, atorvastatin does not alter the PK of edoxaban (Table 7) [34]. The metabolism of rosuvastatin and pravastatin only marginally involve CYP3A4, and Fluvastatin is substrate of CYP2C9 [75]. Thus, the use of pravastatin, Fluvastatin, and rosuvastatin seem to be safer alternatives to simvastatin in patients treated with DOACs.

Fibrates are a second class of lipid lowering agents that might interact with DOAC metabolism. Fenofibrate shows a modest P-gp inhibitory activity in vitro; thus, its interaction with DOACs may not be clinically relevant [79]. Ezetimibe, frequently utilized in combination with statins, does not induce or inhibit CYP3A4 or P-gp, so interactions with DOACs seem to be improbable.

Finally, monoclonal antibodies (mAbs) anti PCSK9, evolocumab and alirocumab, are not metabolized or substrate of CYP and P-gp [80]; thus, no interactions are predicted with DOACs.

### Summary

Although some evidence suggests a possible DDI with simvastatin and lovastatin, these are not considered to have an important clinical impact. No interaction is predicted with ezetimibe and mAbs anti PCSK9.

## 8. Potential Drug–Drug Interaction with Antibiotics and Antifungal Drugs

Some antibiotics and antifungal drugs show moderate to strong inhibition or induction of P-gp (Table 2), with potentially relevant DDI with DOACs that may require a dose adjustment.

The macrolides, clarithromycin and erythromycin, are well-known P-gp and CYP3A4 inhibitors. A slight increase of dabigatran AUC and C_max_ by about 19% and 15%, respectively, has been observed in response to clarithromycin treatment [81]. Clarithromycin and erythromycin also increase rivaroxaban AUC and C_max_ by approximately 40–50% [82]. However, these changes are not considered to be clinically relevant. Similar considerations can be done for apixaban, where the observed increase of 60% of AUC and 30% of C_max_ after co-administration of clarithromycin or erythromycin does not require a dose adjustment [81] (Table 8). On the contrary, the PK of edoxaban seems to be more affected by the co-administration of erythromycin, with a 47% decrease in the total apparent clearance of the drug, associated to a significant increase in both peak (+68%) and total exposure (+85%) of edoxaban and its active metabolite M4 [83]. This interaction may be explained by the inhibition of P-gp, which may result in increased bioavailability of edoxaban in the gut by erythromycin [83]. This pharmacological interaction can be managed by dose adjustment [2], in line with the SmPC (Table 7).

The administration of fluconazole 400 mg (given for 4 days) significantly increases rivaroxaban C_max_ AUC and by 28% and 42%, respectively [82]. This interaction becomes even more importantly when to fluconazole was added also ciclosporin (strong P-gp inhibitor) that determined an increase of rivaroxaban average exposure by 86% and C_max_ by 115% [84]. Thus, patients treated with rivaroxaban in combination with multiple modulators of P-gp (cyclosporin) and CYP3A4 (fluconazole) require particular care. The clinical impact of the combination of therapy with DOACs and fluconazole has been recently investigated by using the nationwide Danish registers [85]. This analysis observed that apixaban users had a higher risk of bleeding following exposure to fluconazole (OR 3.5) while no differences were found among rivaroxaban and dabigatran users [85]. Topical azole exposure did not increase bleeding risk with any DOACs [85].

Ketoconazole, a strong P-gp and CYP3A4 inhibitor, increased total exposure of edoxaban by approximately 90% [83]. Ketoconazole, by inhibiting the P-gp, also increased the bioavailability of M4 metabolite by approximately 46%, without altering the formation mediated by carboxylesterase 1(CES-1) [83]. A significant increase of rivaroxaban AUC and C_max_ by 82% and 53%, respectively was observed in response to Ketoconazole 200 mg once daily, with a concomitant 45% reduction of its clearance [82]. Apixaban maximum plasma concentration and AUC increase by 62% and 99%, respectively, with co-administration of ketoconazole [86]. Thus, the dose of edoxaban should be reduced by 50% in case of a co-administration with antifungals (itraconazole, ketoconazole, and posaconazole, Table 8), whereas fluconazole is not expected to alter the PK of edoxaban [2]. Differently, current guidelines contraindicated the use of apixaban and rivaroxaban with itraconazole, ketoconazole, posaconazole, and voriconazole (Table 8).

Rifampin is one of the most potent inducers of CYP3A4/5 and P-gp; for this reason, a clinically significant DDI may be predicted with DOAC, with a potential differentiation in the case of edoxaban. Rifampin reduces by 34% the total exposure to edoxaban (AUC), although a a concomitant compensatory 5- and 4-fold increase of C_max_ values of metabolites M4 and M6 is observed [87]. These results suggest that rifampin reduce oral bioavailability of edoxaban but increase its metabolism to form the metabolite M6 through CYP3A4/5. The increase plasma levels of the active metabolite M4 is potentially due to the inhibition of the hepatic drug transporter OATP1B1 by rifampin, which determines a impaired liver uptake of the metabolite [87,88]. Starting from this evidence, the administration of edoxaban with rifampin is possible but with caution and, alternatively, should be avoided when possible (Table 8) [2]. Apart from edoxaban, other DOACs are contraindicated with rifampin. 

Quinolones, levofloxacin, and ciprofloxacin are CYP1A2 inhibitors and no relevant interactions are predicted with DOACs [72]. 

### Summary

The DDI with antibiotics and antifungal agents are the most important and with clear clinical evidence. Rifampin is a well-known P-gp inducer and clearly affect the exposure of DOACs, with the exception of edoxaban that shows a partial compensatory increase of its active metabolite and may be considered for a co-treatment. All azole antifungal agents, except for fluconazole, show a strong DDI with DOACs, due to their inhibition of P-gp. However, there are still some missing data for voriconazole; thus, unpredictable DDI can be envisioned with dabigatran and edoxaban.

## 9. Potential Drug–Drug Interaction with Antiacid Drugs

The prevalence of gastro-esophageal reflux disease is significant worldwide and increasingly higher portion of the population is using antacid medication [89]. Dabigatran absorption increases in an acid environment, and, for this reason, may be influenced by the coadministration of antiacids, i.e., the proton pomp inhibitors (PPIs). The solubility of edoxaban is also pH dependent, practically insoluble at a basic pH (8 to 9), slightly soluble at neutral pH (pH 6 to 7), and highly soluble in an acidic pH (pH 3 to 5) [90]. Thus, its bioavailability could be reduced with PPIs. Indeed, its oral bioavailability is maximal at lower pH [91]. Several studies demonstrated that PPI co-administrated with dabigatran decreased dabigatran trough and peak plasma levels [11,92,93,94]. For instance, DDI studies with dabigatran, demonstrated that the use of PPIs, such as pantoprazole 40 mg bid, decreases its AUC by 20–30% and the C_max_ by 45% [11,95], while no effect was seen with ranitidine (Table 9) [37]. In agreement with this interaction, a 2-week period of PPI withdrawal leads to a significant increase in dabigatran trough and peak plasma levels in patients with AF [96]. However, this interaction has not been considered clinically relevant [50], but instead the use of PPI reduced the incidence of hospitalization for upper GI tract bleeding of AF patients [97]. Indeed, the use of PPI is recommended in AF patients under treatment with DOACs, by the National Association of Hospital Cardiologists (ANMCO) and the Italian Association of Hospital Gastroenterologists and Endoscopists (AIGO) [98]. 

Finally, esomeprazole did not show any significant changes in the peak and total exposure of edoxaban during concurrent dosing [99], although omeprazole and pantoprazole may inhibit the P-gp (Table 9) [100]. The influence of omeprazole (once daily 40 mg for 5 days) on the PK and PD of a single 20-mg dose of rivaroxaban, has been investigated in healthy subjects [101]. No clinically meaningful interactions were observed, suggesting that rivaroxaban absorption is not influenced by the gastric pH (Table 9) [101]. 

### Summary

Although dabigatran and edoxaban show a pH-dependent GI absorption, there is no evidence for clinically relevant DDI with antiacids. Instead, the use of PPI shows some protection for upper gastrointestinal tract bleeding [97].

## 10. Potential Drug–Drug Interaction with Antineoplastic and Immune-Modulating Agents

Cancer patients are often treated with DOACs for their higher risk of VTE. Large phase III clinical trials on oncologic patients have been completed with both edoxaban (Hokusai VTE Cancer) and apixaban (Caravaggio) [102,103]. 

A clear statement on possible DDI with DOACs is possible only for the classes of drugs with well-defined effect on P-gp and CYP3A4 (Table 10). Among them, the kinase inhibitors strongly affect the P-gp activity; indeed, imatinib and crizotinib are contraindicated with DOACs [2]. Differently, ibrutinib significantly increases risk of AF, with an estimated cumulative incidence of 5.9% and 10.3% at 6 months and 2 years of treatment, respectively [104]. The use of ibrutinib is also complicated by its inhibitory action on P-gp; thus, its combination with DOACs should be limited and used with caution [104]. Other chemotherapeutic drugs may also increase the incidence of AF, such as alkylating agents (e.g., melphalan, cisplatin, and cyclophosphamide (CTX)), cancer targeted therapies (e.g., sorafenib and sunitinib), and anthracycline agents (e.g., doxorubicin). Thus, their combination with DOACs may be considered but with caution. 

A retrospective analysis of the Caravaggio trial was recently conducted to evaluate the possible DDI between apixaban and anticancer agents. Although there was a limited number of patients under anticancer agent treatment and apixaban (*n* = 336), or dalteparin (*n* = 332), the risks of recurrent VTE, major bleeding and death were similar to those observed in patients taking apixaban or dalteparin alone [105]. Nonetheless, the study was underpowered to identify potential differences in VTE recurrence and bleeding with different classes of anticancer drugs. 

Although the PK interaction between DOACs and monoclonal antibodies is not expected [80], some of these new target therapies may increase the risk of bleeding and thus their combination with anticoagulants should be avoided or conducted with caution. This PD interaction can be predicted for the anti CD52 alemtuzumab, contraindicated with all DOACs, while bevacizumab (anti VEGF), caplacizumab (anti-vWF), ipilimumab (anti CTLA4) and ramucirumab (anti-VEGFR2) could be used with caution in the presence of DOACs (Table 10). 

Considering the hormone therapies, enzalutamide, an androgen receptors antagonist, is a strong inducer of P-gp and CYP3A4 and a moderate inducer of CYP2C9 and CYP2C19 [106,107]. For these pharmacological properties its use is contraindicated with all DOACs (Table 10). Indeed, patients with prostate cancer associated VTE can be safely treated with low molecular weight heparin (LMWH) while on enzalutamide, though observations of real-world use suggest alternative oral anticoagulants are used just as frequently [108]. The androgen receptor antagonist tamoxifen is predicted to have a lower impact on the PK of DOACs and can be used with caution (Table 10). Results from a recent retrospective study conducted in breast cancer women with AF, suggests that DOACs are an effective and safe therapeutic option during adjuvant hormonal therapy with either tamoxifen or aromatase inhibitors [109].

Among the immune-modulating agents, the majority of the studies detecting a possible interaction with DOACs have been conducted with cyclosporine and tacrolimus. Several in vitro studies identified cyclosporine, tacrolimus, and rapamycin as inhibitors of CYP3A4 and Pgp. More importantly, by using a validated intravenous and per oral ^14^C erythromycin breath and urine test, Lemahieua, et al. observed a significant increase in intestinal CYP3A4 activity and a significant decrease in hepatic and intestinal P-gp activity in patients on cyclosporine in comparison with those on tacrolimus and rapamycin [110]. This data suggested a significant interaction of cyclosporine with all DOACs, although the strongest effect has been observed with dabigatran [37] (contraindicated) and with edoxaban [83] (+1.7 fold of Cmax and AUC of edoxaban + cyclosporine vs. edoxaban alone). On the contrary, cyclosporine and tacrolimus showed a neglecting effect on the PK of apixaban in healthy volunteers [111]. Finally, in healthy volunteers, cyclosporine increased the exposure of rivaroxaban by 46% and the Cmax by more than 2-fold [84]. The clinical impact of this potential DDI has been also evaluated in kidney transplanted patients treated with warfarin or DOACs in the presence of tacrolimus or cyclosporine [112]. This retrospective analysis revealed that the rates of major bleeding were 7.2% per year with DOACs vs. 11.4% per year with warfarin and that the lowest incidence was observed with apixaban compared to all other anticoagulants (6.7% vs. 19.0%), further supporting the lower incidence of immunosuppressor agents on apixaban exposure and activity [112]. 

Dexamethasone, prednisone, and other corticosteroids may increase the bleeding risk at upper GI level; thus, a potential PD interaction is predicted (Table 10). In addition, dexamethasone may induce P-gp and partially affect the anticoagulant action of DOACs, although clinical data suggesting that dexamethasone is a P-gp inducer are limited and indirect [113,114].

### Summary

The majority of the chemotherapeutic agents appear to be neglected of DDI with DOACs. The tyrosine kinase inhibitors certainly represent the most critical class of drugs with clinically relevant DDI with DOACs due to their inhibition of P-gp. However, this DDI are restricted mainly to imatinib and crizotinib. Alemtuzumab increases the bleeding risk as monotherapy, thus a PD interaction can be envisioned with DOACs. Finally, enzalutamide has the peculiarity to inhibit P-gp and thus its use is contraindicated with all DOACs. Finally, dabigatran and edoxaban show a more relevant DDI with cyclosporine and tacrolimus, immunosuppressant agents with strong to moderate P-gp inhibitory effect. 

## 11. Potential Drug–Drug Interaction with Antiepileptic Agents

Stroke accounts for 30–40% of all cases of epilepsy in the elderly [115]. These patients may require anticoagulant therapy and it is predicted to receive a concomitant therapy with antiepileptic drugs. 

Although there are few clinical evidences of DDI between DOACs and antiepileptic drugs, in vitro studies clearly documented that many of these drugs induce CYP3A4 and P-gp leading to reduced DOACs exposure (Table 11) [116]. For instance, carbamazepine, phenobarbital, and phenytoin are potent inducers of P-gp [117], and therefore may lead to reduced DOACs plasma concentrations and clinical efficacy. According to the EHRA practical guide, the use of carbamazepine, phenobarbital, and phenytoin is only possible with edoxaban and apixaban [2]. Indeed, the use of inducing P-gp agents may decrease absorption of all DOACs and may reduce their efficacy. The higher variability of plasma concentration of rivaroxaban and dabigatran (peak/through ratio) [118] may have a relevant clinical impact on their interaction with inducing agents. In addition, the influence of P-gp inducers on edoxaban can be considered less problematic due to the compensatory increase of the active metabolite M4, as reported for rifampicin [87]. 

A careful monitoring of the efficacy of all DOACs is, instead, indicated for valproic acid, due to its more potent modulation of P-gp [50]. Clinical data demonstrated, by using digoxin as P-gp substrate, that levetiracetam does not induce P-gp and thus can be utilized with DOACs, although with caution [119,120]. The antiepileptic drugs that do not affect P-gp function, and thus are not predicted to interact with DOACs include ethosuximide, lamotrigine, gabapentin, oxcarbazepine, pregabalin, topiramate, and zonisamide [50]. 

### Summary

Antiepileptic drugs, such as carbamazepine, phenobarbital, and phenytoin, are well-known CYP450 and P-gp inducers and their use is contraindicated for dabigatran and rivaroxaban, the two DOACs with higher variability in their plasma concentrations. For apixaban and edoxaban, there is no direct evidence of a DDI with these antiepileptic drugs and thus may be co-administered with caution, and a lower efficacy might be predicted. Moreover, valproic acid induces the P-gp activity, although at lower extent; thus, DOACs may be co-administered but with caution. Finally, levetiracetam has been re-classified from the previous EHRA guidelines and it is not predicted to have a clinically significant DDI with DOACs, since its effect on P-gp is neutral [120].

## 12. Potential Drug–Drug Interaction with Antiviral Agents for Human Immunodeficiency and Hepatitis C Viruses

Considering the HIV therapy, darunavir boosted with ritonavir or cobicistat is the main protease inhibitor still recommended as first-line treatment. The majority of the HIV protease inhibitors strongly affect the CYP3A4 activity, with ritonavir being the most potent, saquinavir the least and tipranavir with no effect. However, ritonavir is also a strong P-gp inhibitor and thus it is expected to increase DOACs exposure [121]. Therefore, the co-administration of darunavir/ritonavir with DOACs is not recommended [2] (Table 12). Similarly, the pharmacoenhancer cobicistat, a potent CYP3A4, P-gp, and BCRP inhibitor [122], is predicted to increase DOACs bioavailability; thus, the combinations atazanavir/cobicistat and darunavir/cobicistat are contraindicated in patients under DOAC treatment.

HCV-infected patients are at higher risk of VTE, coupled with an increased incidence of AF [123], thus resulting in a DOAC prescription. In this context, the use of DOACs in patients with cirrhosis is controversial due to possible new onset of liver injury [124]. The interactions between dabigatran and glecaprevir/pibrentasvir [125] or sofosbuvir/velpatasvir/voxilaprevir [126] has been recently investigated in specific phase I trials. Dabigatran AUC increases by 138%, and 161%, when co-administered with glecaprevir/pibrentasvir and sofosbuvir/velpatasvir/voxilaprevir, respectively (Table 13).

Paritaprevir is an HCV protease inhibitor that is boosted with ritonavir and thus this combination is predicted to increase the exposure of DOACs. 

In vitro data indicate that Grazoprevir is not a P-gp inhibitor, and thus it is not expected to interact with DOACs [127]. Sofosbuvir, a nonstructural protein 5AB (NS5B) polymerase inhibitor, is not metabolized by CYP450 although it is a P-gp substrate (Table 13) [128]. Thus, the fixed-dose combination of sofosbuvir and ledipasvir, a substrate and inhibitor of P-gp/BCRP, may have a relevant impact on DOACs PK and could increase their exposure (Table 13) [129].

### Summary

HIV protease inhibitor, primarily if enhanced with ritonavir and cobicistat, are potent inhibitors of CYP3A4 and P-gp; thus, their use is contraindicated with DOACs. A possible interaction is also predicted with anti HCV therapies, such as with NS5B inhibitor sofosbuvir in combination with velpatasvir/voxilaprevir and with the fixed combination of glecaprevir/pibrentasvir.

## 13. Potential Drug–Drug Interaction with Anti-COVID-19 Agents

Remdesivir, darunavir, hydroxychloroquine, atazanavir, lopinavir, and interferon beta-1a are and have been the recommended therapy for coronavirus disease 2019 (COVID-19). Lopinavir, darunavir, atazanavir, and nirmatrelvir are all utilized in combination with ritonavir as booster and, for this reason, are all considered contraindicated with DOACs (Table 14). As discussed before, methylprednisolone and other corticosteroids may also interfere with DOACs mainly by an increase of GI bleeding risk [85,114]. 

### Summary

For the anti-COVID-19 therapy, ritonavir is still the drug with higher impact on the PK of DOACs. 

## 14. Potential Drug–Drug Interaction with Monoclonal Antibodies Anti Interleukin 6

DDI between therapeutic monoclonal antibodies (mAbs) and DOACs is unlikely due to the fact that their clearance and distribution does not involve CYP450 and P-gp [80]. The only exception is represented by the mAbs anti interleukin (IL)-6, such as tocilizumab. Indeed, IL-6 downregulates the CYP3A4, 2C8 and 2B6 mRNA expression, and IL-6 reduced by 70% P-gp expression in mice. Thus, the inhibition of IL-6 by tocilizumab, induced P-gp and reducing DOAC bioavailability (Table 14). A DDI between tocilizumab and dabigatran has been described with a progressively decreased anticoagulant effect, favoring mesenteric arterial thrombosis [130]. The coadministration of dabigatran with tocilizumab induced similar interaction can be predicted for the other DOACs. For dupilumab, a mAb directed anti IL-4 and IL-13 signaling, may be predicted a similar effect. An open-label DDI study was performed to assess the interaction of dupilumab with the PK of drugs metabolized by CYP450 enzymes, including warfarin [131]. The results clearly show no significant DDI of drugs metabolized by CYP3A, CYP2C19, CYP2C9, CYP1A2, and CYP2D6 by dupilumab [131].

### Summary

The mAb anti IL-6 receptor, tocilizumab, shows a very peculiar effect compared to other mAbs. Indeed, tocilizumab may determine a significant induction of P-gp and CYP3A4 with possible exposure of patients to thrombosis by affecting the GI absorption of DOACs. 

## 15. Expert Opinion

DDIs with DOACs are receiving recent attention in the scientific and health care communities due to the fact that AF or VTE are commonly associated with many other cardiac and non-cardiac chronic conditions, including cancer, hematological and immunological disorders [132]. One direct consequence of multimorbidity is polypharmacy, which identifies the utilization of multiple drugs, increasing the chance of experiencing DDIs, which are common causes of adverse drug reactions. In this complex scenario, it must be also considered that a large number of drugs are introduced every year, and new interactions between medications are increasingly reported. Nevertheless, the introduction of the new class of direct anticoagulant has potentially simplified the treatment with multiple drugs considering that a total of 605 drugs are known to interact with warfarin, categorized as 136 major, 398 moderate, and 71 minor interactions. However, dabigatran, apixaban, rivaroxaban, and edoxaban cannot be commonly considered when co-prescribed with other drugs. Indeed, their peculiar PD (dabigatran vs. anti fXa) and PK properties must be considered for a proper prediction of clinically relevant DDIs. Indeed, although all DOACs are substrate of the P-gp and may interact with strong inhibitors and inducers of this drug transporter, the inter-individual variability of drug plasma concentrations, lower for apixaban and edoxaban and higher for rivaroxaban and dabigatran, is a determining factor for triggering a clinically significant DDIs. Secondly, the effect of perpetrators (CYP3A4 inhibitors or inducers) on DOACs exposure is predicted to be more relevant for rivaroxaban and apixaban which undergo to CYP-mediated metabolism. An additional aspect that requires further investigation for a better prediction of DDI with DOACs is represented by the analysis of possible associations between genetic variants and PK profile. This topic was not covered in the present review but some important information can be found in a recent systematic review [133]. Finally, the anti-fXa chromogenic assays are available to measure plasma concentrations of DOACs. This determination may help clinicians to detect a DDIs [2].

## 16. Conclusions

In conclusion, although the majority of the possible interactions are predicted and not studied in specific clinical trials, in response to anticipated DDIs, possible strategies, including dosage reduction or different time of administrations, are recommended. Nevertheless, additional clinical PK studies and analysis of registry, as anticipated by a retrospective cohort study using data from the Taiwan National Health Insurance database [40], will be necessary to ascertain the DDIs, which are currently mainly derived from hypothetical conclusions.

## Figures and Tables

**Table 1 pharmaceutics-14-01120-t001:** Pharmacokinetic and pharmacodynamic characteristics of DOACs.

	Dabigatran	Rivaroxaban	Apixaban	Edoxaban
Target	Thrombin	fXa	fXa	fXa
Ki (nmol/L)	4.5	0.4	0.08	0.56
Bioavailability	6.5% (absolute)	80% (absolute)	66% (absolute)	60% (absolute)
Effect of food	Delayed and not reduced absorption	Increased absorption (20 mg)	None	None
Administered with food	No	Yes *	No	No
Vd (L)	60–70	50	21	>300
Protein bound	35%	>90%	87%	40–59%
prodrug	Yes	No	No	No
Tmax (h)	1–3	2–4	3–4	2
Peak levels (ng/mL) **	175 (117–275)	249 (184–343)	171 (91–321)	170 (125–245)
Trough levels (ng/mL) **	91 (61–143)	44 (12–137)	103 (41–230)	36 (16–62)
Half-life (h)	12–17	5–9 (healthy)	8–15	8–11
Metabolism (CYP)	Conjugation	3A4 (18%), 2J2, and CYP independent	3A4 (25%), 1A2, 2J2, 2C8, 2C9, 2C19	3A4 (<4%)
P-gp substrate	Yes (only prodrug)	Yes	Yes	Yes
Substrate of other transporters	Not known	BCRP/ABCG2	BCRP/ABCG2	Not known
Renal elimination	80%	35%	27%	50%
Hemodialysis elimination	60–70%	Unlikely	Unlikely	Unlikely
Administration frequency	Double daily dose	Once daily dose	Double daily dose	Once daily dose

Vd: volume of distribution; Tmax: Time to reach the maximal plasma concentration; CYP450: Cytochrome P 450; P-gp. P-glycoprotein. * The drug at 15 and 20 mg must be administered with food; ** Expected plasma levels of DOACs in patients treated for AF.

**Table 3 pharmaceutics-14-01120-t003:** Effects of cardiovascular drugs on DOACs exposure.

Concomitant Drug		Effect on DOACs Concentration			
Cardiovascular Drugs	Effect on P-gp and CYP	Dabigatran	Rivaroxaban	Apixaban	Edoxaban
Amiodarone	Moderate P-gp competition	●+12 to 60%	●Minor effect	●Modest increase of concentrations	●+40% AUC
Digoxin	P-gp competition	No effect	No effect	No effect	No effect
Diltiazem	P-gp competition and weak CYP3A4 inhibition	Possible increase of concentrations	●Possible increase of concentrations	●Increase in AUC (1.4-fold) and C_max_ (1.3-fold)	●No significant effect on AUC predicted
Dronedarone	Moderate P-gp inhibition and CYP3A4 inhibition	● +70 to 100%	● Increase in bleeding risk (+30–40%)	●Possible increase of concentrations	●+85% AUC
Quinidine	P-gp competition	●+53% AUC	●Extent in increase unknown	●Extent in increase unknown	●+77% AUC (no dose reduction required by label)
Verapamil	Moderate P-gp inhibition and weak CYP3A4 inhibition	●+12 to 180% AUC (reduce to 10 mg bid)	●+40% AUC Increase in bleeding risk	●Extent in increase unknown	●+53% AUC (no dose reduction required by label)
Atenolol	P-gp substrate	No PK data	No PK data	AUC and Cmax unchanged	No PK data

AUC = Area under the curve; CYP = Cytochrome P 450; P-gp = P-glycoprotein. White: No relevant DDI anticipated. Yellow: caution/careful monitoring required, especially in case of polypharmacy or in the presence of ≥2 yellow/bleeding risk factors. Orange: Consider dose reduction or avoiding concomitant use. Red: Contraindicated/not advisable. Blue dot indicates PK interaction.

**Table 4 pharmaceutics-14-01120-t004:** Effects of antiplatelet and antithrombotic drugs on DOAC exposure and pharmacological activity.

Concomitant Drug		Effect on DOACs Concentration and Pharmacodynamic			
**Antiplatelet Drugs**	**Effect on P-gp and CYP**	**Dabigatran**	**Rivaroxaban**	**Apixaban**	**Edoxaban**
Clopidogrel	No relevant PK interactions known/assumed	●●+30–40% AUC and C_max_; Pharmacodynamically increased bleeding time	●●No significant effect on AUC predicted; Pharmacodynamically increased bleeding time	●●No significant effect on AUC predicted; Pharmacodynamically increased bleeding time	●●No significant effect on AUC predicted; Pharmacodynamically increased bleeding time
Ticagrelor	P-gp inhibition	●●+25–70% AUC; Pharmacodynamically increased bleeding time	●No data Pharmacodynamically increased bleeding time	●No data Pharmacodynamically increased bleeding time	●Predicted increase of AUC; Pharmacodynamically increased bleeding time
Aspirin	No relevant effect known/assumed	●Pharmacodynamically increased bleeding time			●●Increased AUC for high doses of aspirin; Pharmacodynamically increased bleeding time
Prasugrel	P-gp substrate	●Pharmacodynamically increased bleeding time			●No significant effect on AUC; Pharmacodynamically increased bleeding time
Cilostazol, Dipyridamole	No relevant effect known/assumed	●Pharmacodynamically increased bleeding time			
**Prostacyclin Analogues**	**Effect on P-gp and CYP**	**Dabigatran**	**Rivaroxaban**	**Apixaban**	**Edoxaban**
Epoprostenol, Iloprost, Treprostinil	No relevant effect known/assumed	●Pharmacodynamically increased bleeding time			

AUC = Area under the curve; CYP = Cytochrome P 450; P-gp = P-glycoprotein. White: No relevant DDI anticipated. Yellow: caution/careful monitoring required, especially in case of polypharmacy or in the presence of ≥2 yellow/bleeding risk factors. Orange: Consider dose reduction or avoiding concomitant use. Red: Contraindicated/not advisable. Blue dot indicates PK interaction and violet dot PD interaction.

**Table 5 pharmaceutics-14-01120-t005:** Effects of NSAIDs on DOAC exposure and pharmacological activity.

Concomitant Drug		Effect on DOACs Concentration and Pharmacodynamic			
NSAIDs	Effect on P-gp and CYP	Dabigatran	Rivaroxaban	Apixaban	Edoxaban
Naproxene	P-gp competition; CYP1A2 and CYP2C9 inhibition	●No data; Pharmacodynamically increased bleeding time		●●+55% AUC; Pharmacodynamically increased bleeding time	●No PK effect; Pharmacodynamically increased bleeding time
Other NSAIDs	No relevant PK interactions known/assumed	●Pharmacodynamically increased bleeding time			

AUC = Area under the curve; CYP = Cytochrome P 450; P-gp = P-glycoprotein. Orange: Consider dose reduction or avoiding concomitant use. Blue dot indicates PK interaction and violet dot PD interaction.

**Table 6 pharmaceutics-14-01120-t006:** Effects of antidepressant on DOAC exposure and pharmacological activity.

Concomitant Drug		Effect on DOACs Concentration and Pharmacodynamic			
Antidepressant	Effect on P-gp and CYP	Dabigatran	Rivaroxaban	Apixaban	Edoxaban
St. John’s wort (*Hypericum perforatum* L.)	Strong CYP3A4 and P-gp induction	● Relevant decrease in AUC predicted	● −24% AUC and −14% Cmax	● Relevant decrease in AUC predicted	● Relevant decrease in AUC predicted
SSRI	No relevant PK interactions known/assumed; Fluvoxamine is a mild inhibitor of CYP3A4	●Pharmacodynamically increased bleeding risk			
Clomipramine	No relevant PK interactions known/assumed	●Pharmacodynamically increased bleeding risk			
Vortioxetine	No relevant PK interactions known/assumed	●Pharmacodynamically increased bleeding risk			

AUC = Area under the curve; CYP = Cytochrome P 450; P-gp = P-glycoprotein. Yellow: caution/careful monitoring required, especially in case of polypharmacy or in the presence of ≥2 yellow/bleeding risk factors. Orange: Consider dose reduction or avoiding concomitant use. Red: Contraindicated/not advisable. Blue dot indicates PK interaction and violet dot PD interaction.

**Table 7 pharmaceutics-14-01120-t007:** Effects of lipid-lowering drugs on DOAC exposure.

Concomitant Drug		Effect on DOACs Concentration			
Lipid-Lowering Drug	Effect on P-gp and CYP	Dabigatran	Rivaroxaban	Apixaban	Edoxaban
Atorvastatin	P-gp and CYP3A4 competition	No PK interaction	No effect	No data	+1.7% AUC
					−14.2% Cmax
Simvastatin; Lovastatin	P-gp moderate inhibition; CYP3A4 substrate	●Possible increased exposure	No data	No data	No data
			Minor effect on AUC predicted	Minor effect on AUC predicted	Minor effect on AUC predicted
Fluvastatin	CYP2C9 substrate	No significant effect on AUC predicted			
Fenofibrate	P-gp inhibitor	Minor effect on AUC predicted			
Gemfibrozil	CYP2C8 inhibitor	No significant effect on AUC predicted			
Ezetimibe	No relevant PK interactions known/assumed	No data, no significant effect on AUC predicted;			
PCSK9 inhibitors	No relevant PK interactions known/assumed	No data, no significant effect on AUC predicted;			

AUC = Area under the curve; CYP = Cytochrome P 450; P-gp = P-glycoprotein. White: No relevant DDI anticipated. Yellow: caution/careful monitoring required, especially in case of polypharmacy or in the presence of ≥2 yellow/bleeding risk factors. Blue dot indicates PK interaction.

**Table 8 pharmaceutics-14-01120-t008:** Effects of antibiotics and antifungal drugs on DOACs exposure and pharmacological activity.

Concomitant Drug		Effect on DOACs Concentration and Pharmacodynamic			
**Antibiotics**	**Effect on P-gp and CYP**	**Dabigatran**	**Rivaroxaban**	**Apixaban**	**Edoxaban**
Erythromycin	P-gp substrate; CYP3A4 inhibition	●Predicted +15 to 20% AUC	●+34% AUC	●Predicted +60% AUC +30% C_max_	●+85% AUC
Clarithromycin	P-gp and CYP3A4 inhibition	●+15 to 100% AUC	●+54% AUC +40% C_max_	●+60% AUC +30% C_max_	●Predicted increase of AUC
Rifampin	P-gp/ BCRP and CYP3A4/CYP2J2 induction	● − 66% AUC	● − 50% AUC	● − 54% AUC	●AUC: −35%, compensatory increase of active metabolites
Metronidazole	CYP3A4 inhibition	No significant effect on AUC predicted			
Levofloxacin Ciprofloxacin	CYP1A2 inhibition	No significant effect on AUC predicted			
Cephazolin	No relevant PK interactions known/assumed	●Pharmacodynamically increased bleeding time			
**Antifungals**	**Effect on P-gp and CYP**	**Dabigatran**	**Rivaroxaban**	**Apixaban**	**Edoxaban**
Fluconazole	Moderate CYP3A4 inhibition	●Predicted AUC increase	●+42% AUC	●Predicted AUC increase	No data
Ketoconazole, itraconazole	Potent P-gp and BCRP competition; CYP3A4 inhibition	● +140 to 150% AUC	● Up to 160% AUC	● +100% AUC	●+87 to 95% AUC
Posaconazole	Potent P-gp competition; CYP3A4 inhibition	●Predicted increase of AUC	● Predicted up to +100% AUC	● Predicted up to +100% AUC	●Predicted increase of AUC
Voriconazole	Potent CYP3A4 inhibition	No data	● Predicted up to +100% AUC	● Predicted up to +100% AUC	No data

AUC = Area under the curve; CYP = Cytochrome P 450; P-gp = P-glycoprotein. White: No relevant DDI anticipated. Yellow: caution/careful monitoring required, especially in case of polypharmacy or in the presence of ≥2 yellow/bleeding risk factors. Orange: Consider dose reduction or avoiding concomitant use. Red: Contraindicated/not advisable. Blue dot indicates PK interaction and violet dot PD interaction.

**Table 9 pharmaceutics-14-01120-t009:** Effects of antiacid drugs on DOACs exposure.

Concomitant Drug		Effect on DOACs Concentration			
PPI	Effect on P-gp and CYP	Dabigatran	Rivaroxaban	Apixaban	Edoxaban
Pantoprazole	GI absorption P-gp and CYP2C9 inhibition	●−20–30% AUC −45% Cmax	No data	No data	No data
Esomeprazole	GI absorption	No data	No data	No data	No significant effect
Omeprazole	GI absorption P-gp and CYP2C9 inhibition	No data	No significant effect	No data	No data
Ranitidine	GI absorption	No effect	No data	No data	No data
Aluminum-Magnesium Hydroxide	GI absorption	No data	No data	No data	No data

AUC = Area under the curve; CYP = Cytochrome P 450; P-gp = P-glycoprotein. White: No relevant DDI anticipated. Blue dot indicates PK interaction.

**Table 10 pharmaceutics-14-01120-t010:** Effects of antineoplastic drugs on DOACs exposure and pharmacological activity.

Concomitant Drug		Effect on DOACs Concentration and Pharmacodynamic Effect			
**Antimitotic Agents**	**Effect on P-gp and CYP**	**Dabigatran**	**Rivaroxaban**	**Apixaban**	**Edoxaban**
Paclitaxel	Moderate CYP3A4 induction; CYP3A4/P-gp competition	No significant effect on AUC predicted			
Vinblastine, Vincristine, Vinca alkaloids	CYP3A4/P-gp competition	●Mild decrease in AUC predicted			
Docetaxel	Mild CYP3A4 induction; CYP3A4/P-gp competition	No significant effect on AUC predicted			
**Antimetabolites**	**Effect on P-gp and CYP**	**Dabigatran**	**Rivaroxaban**	**Apixaban**	**Edoxaban**
Metotrexate	P-gp competition; no relevant interaction anticipated	No significant effect on AUC predicted			
Pemetrexed, Purine analogs, Pyrimidine analogs	No relevant interaction anticipated	No significant effect on AUC predicted			
**Topoisomerase inhibitors**	**Effect on P-gp and CYP**	**Dabigatran**	**Rivaroxaban**	**Apixaban**	**Edoxaban**
Topotecan	No relevant interaction anticipated	No significant effect on AUC predicted			
Irinotecan	CYP3A4/P-gp competition; no relevant interaction anticipated	No significant effect on AUC predicted			
Etoposide	Mild CYP3A4 induction; CYP3A4/P-gp competition	No significant effect on AUC predicted			
**Anthracyclines/** **Anthracenediones**	**Effect on P-gp and CYP**	**Dabigatran**	**Rivaroxaban**	**Apixaban**	**Edoxaban**
Doxorubicin	CYP3A4/P-gp competition	●Decrease in AUC predicted			
Idarubicin	Mild CYP3A4 inhibition; P-gp competition	No significant effect on AUC predicted			
Daunorubicin	P-gp competition; no relevant interaction anticipated	No significant effect on AUC predicted			
Mitoxantrone	No relevant interaction anticipated	No significant effect on AUC predicted			
**Alkylating agents**	**Effect on P-gp and CYP**	**Dabigatran**	**Rivaroxaban**	**Apixaban**	**Edoxaban**
Ifosfamide	Mild CYP3A4 inhibition; CYP3A4 competition	No significant effect on AUC predicted			
Ciclophosphamide	Mild CYP3A4 inhibition; CYP3A4 competition	No significant effect on AUC predicted			
Lomustine	Mild CYP3A4 inhibition	No significant effect on AUC predicted			
Busulfan	CYP3A4 competition; no relevant interaction anticipated	No significant effect on AUC predicted			
Bendamustine	P-gp competition; no relevant interaction anticipated	No significant effect on AUC predicted			
Chlorambucil, Melphalan, Carmustine, Procarbazine, Dacarbazine, Temozolomide	No relevant effect anticipated	No significant effect on AUC predicted			
**Platinum-based agents**	**Effect on P-gp and CYP**	**Dabigatran**	**Rivaroxaban**	**Apixaban**	**Edoxaban**
Cisplatin, Carboplatin, Oxaliplatin	No relevant effect anticipated	No significant effect on AUC predicted			
**Intercalating agents**	**Effect on P-gp and CYP**	**Dabigatran**	**Rivaroxaban**	**Apixaban**	**Edoxaban**
Bleomycin, Dactinomycin	No relevant effect anticipated	No significant effect on AUC predicted			
Mitomycin C	No relevant interaction anticipated	No significant effect on AUC predicted			
**Enzymes**	**Effect on P-gp and CYP**	**Dabigatran**	**Rivaroxaban**	**Apixaban**	**Edoxaban**
Asparaginase, Pegaspargase	No relevant PK interactions known/assumed	●Pharmacodynamically increased bleeding time			
**Tyrosine kinase inhibitors**	**Effect on P-gp and CYP**	**Dabigatran**	**Rivaroxaban**	**Apixaban**	**Edoxaban**
Imatinib, Crizotinib	Strong P-gp inhibition; Moderate CYP3A4 inhibition; CYP3A4/P-gp competition	● Significant increase in AUC predicted			
Tucatinib	Moderate to strong CYP3A4 and P-gp inhibition	●Moderate increase in AUC predicted			
Nilotinib, Lapatinib	Moderate-to-strong P-gp inhibition; mild CYP3A4 inhibition; CYP3A4/P-gp competition	●Possible increase in AUC predicted			
Ribociclib	Moderate to strong CYP3A4 inhibition; CYP3A4/P-gp competition	No significant effect on AUC predicted	●Possible increase in AUC predicted	●Possible increase in AUC predicted	No significant effect on AUC predicted
Vemurafenib	Moderate CYP3A4 induction; P-gp inhibition	●Moderate increase in AUC predicted	●Possible variation in AUC predicted	●Possible variation in AUC predicted	●Possible variation in AUC predicted
Lorlatinib	Moderate CYP3A4 and P-gp induction	●Possible reduction in AUC predicted			
Ceritinib	Strong CYP3A4 inhibition; CYP3A4 and P-gp competition	●Possible increase in AUC predicted			
Selpercatinib	Mild CYP3A4 inhibition; CYP3A4/P-gp competition	●Possible increase in AUC predicted			
Dasatinib	Mild CYP3A4 inhibition; CYP3A4/P-gp competition	●Possible increase in AUC predicted ●Pharmacodynamically increased bleeding risk			
Encorafenib	CYP3A4 competition	●Pharmacodynamically increased bleeding risk			
Vandetanib, Cabozantinib, Neratinib, Osimertinib, Ruxolitinib	P-gp inhibition; CYP3A4 competition	●Possible increase in AUC predicted			
Alectinib, Alpelisib, Brigatinib, Gilteritinib, Pemigatinib	P-gp inhibition	●Possible increase in AUC predicted			
Sunitinib, Avapritinib, Carfilzomib, Glasdegib, Ponatinib	P-gp inhibition; CYP3A4 competition	●Possible increase in AUC predicted ●Pharmacodynamically increased bleeding risk			
Nintedanib	P-gp competition	●Pharmacodynamically increased bleeding risk			
Erlotinib, Gefitinib, Afatinib	CYP3A4 competition, no relevant interaction anticipated	No significant effect on AUC predicted			No PK interaction
Binimetinib	No relevant PK interactions known/assumed	●Pharmacodynamically increased bleeding risk			
Ibrutinib	P-gp inhibition; CYP3A4 competition	●Possible increase in AUC predicted ●Pharmacodynamically increased bleeding risk			
Acalabrutinib, zanubrutinib	CYP3A4 and P-gp competition	●Pharmacodynamically increased bleeding risk			
**BCL-2 inhibitors**	**Effect on P-gp and CYP**	**Dabigatran**	**Rivaroxaban**	**Apixaban**	**Edoxaban**
Venetoclax	P-gp inhibition; CYP3A4 and P-gp competition	●Possible increase in AUC predicted			
**Monoclonal** **antibodies**	**Effect on P-gp and CYP**	**Dabigatran**	**Rivaroxaban**	**Apixaban**	**Edoxaban**
Brentuximab	No relevant interactions anticipated	No significant effect on AUC predicted			
Rituximab, Cetuximab, Trastuzumab	No relevant effect assumed	No significant effect on AUC predicted			
Alemtuzumab	No relevant PK interactions known/assumed	● Pharmacodynamically increased bleeding risk			
Bevacizumab, Caplacizumab, Ipilimumab, Ramucirumab	No relevant PK interactions known/assumed	●Pharmacodynamically increased bleeding risk			
**Hormonal agents**	**Effect on P-gp and CYP**	**Dabigatran**	**Rivaroxaban**	**Apixaban**	**Edoxaban**
Abiraterone	Moderate CYP3A4 inhibition; Strong P-gp inhibition; CYP3A4/P-gp competition	●Possible increase in AUC predicted			
Enzalutamide	Strong CYP3A4 induction; P-gp inhibition; CYP3A4/P-gp competition	● Possible variation in AUC predicted	● Significant decrease in AUC predicted	● Significant decrease in AUC predicted	● Possible variation in AUC predicted
Bicalutamide	Moderate CYP3A4 inhibition	No significant effect on AUC predicted	●Possible increase in AUC predicted	●Possible increase in AUC predicted	No significant effect on AUC predicted
Tamoxifen	Strong P-gp inhibition; Mild CYP3A4 inhibition; CYP3A4 competition	●Moderate increase in AUC predicted			
Anastrozole	Mild CYP3A4 inhibition	No significant effect on AUC predicted			
Flutamide	CYP3A4 competition; No relevant interactions anticipated	No significant effect on AUC predicted			
Letrozole, Fulvestrant	CYP3A4 competition; No relevant interactions anticipated	No significant effect on AUC predicted			
Raloxifene, Leuprolide, Mitotane	No relevant interactions anticipated	No significant effect on AUC predicted			
**Immune-modulating-agents**	**Effect on P-gp and CYP**	**Dabigatran**	**Rivaroxaban**	**Apixaban**	**Edoxaban**
Cyclosporine	Strong to moderate P-gp inhibition, moderate CYP3A4 inhibition; CYP3A4/P-gp competition	● Strong increase of AUC predicted	●+46% AUC +2 fold Cmax	●+ 20% AUC +40% C_max_	●+73% AUC (reduce to 30 mg as indicated by label)
Tacrolimus	Strong to moderate P-gp inhibition, mild CYP3A4 inhibition; CYP3A4/P-gp competition	● Strong increase of AUC predicted	●Possible increase in AUC predicted	●Possible increase in AUC predicted	●Moderate increase in AUC predicted, consider a dose reduction
Dexamethasone	Strong CYP3A4/P-gp induction; CYP3A4/P-gp competition	●Possible decrease in AUC predicted ●Pharmacodynamically increased bleeding risk			
Prednisone and other corticosteroids	Moderate CYP3A4 induction; CYP3A4 competition	●No significant effect on AUC predicted ●Pharmacodynamically increased bleeding risk			
Temsirolimus, Sirolimus	Mild CYP3A4 inhibition; CYP3A4/P-gp competition	No significant effect on AUC predicted			
Everolimus	CYP3A4 competition; No relevant interactions anticipated	No significant effect on AUC predicted			

AUC = Area under the curve; CYP = Cytochrome P 450; P-gp = P-glycoprotein. White: No relevant DDI anticipated. Yellow: caution/careful monitoring required, especially in case of polypharmacy or in the presence of ≥2 yellow/bleeding risk factors. Orange: Consider dose reduction or avoiding concomitant use. Red: Contraindicated/not advisable. Blue dot indicates PK interaction and violet dot PD interaction.

**Table 11 pharmaceutics-14-01120-t011:** Predicted effects of antiepileptic drugs on DOACs exposure.

Concomitant Drug		Effect on DOACs Concentration			
Antiepileptic Drugs	Effect on P-gp and CYP	Dabigatran	Rivaroxaban	Apixaban	Edoxaban
Carbamazepine	Strong CYP3A4/P-gp induction; CYP3A4 competition	● Strong decrease in AUC	● Strong decrease in AUC	●Possible decrease in AUC predicted	●Possible decrease in AUC predicted
Ethosuximide	CYP3A4 competition; No relevant interaction known/assumed	No significant effect on AUC predicted			
Gabapentin	No relevant interactions known/assumed	No significant effect on AUC predicted			
Lamotrigine	P-gp competition; No relevant interaction known/assumed	No significant effect on AUC predicted			
Levetiracetam	P-gp induction; P-gp competition	●Possible decrease in AUC predicted			
Oxcarbazepine	CYP3A4 induction; P-gp competition	No significant effect on AUC predicted			
Phenobarbital	Strong CYP3A4/P-gp induction; P-gp competition	● Decrease in AUC	● Decrease in AUC	●Possible decrease in AUC	●Possible decrease in AUC
Phenytoin	Strong CYP3A4/P-gp induction; P-gp competition	● Decrease in AUC	● Decrease in AUC	●Possible decrease in AUC	●Possible decrease in AUC
Valproic acid	CYP3A4/P-gp induction	●Possible decrease in AUC predicted			
Pregabalin	No relevant interactions known/assumed	No significant effect on AUC predicted			
Topiramate	CYP3A4 induction; CYP3A4 competition	No significant effect on AUC predicted			
Zonisamide	CYP3A4 competition; No relevant interactions known/assumed	No significant effect on AUC predicted			

AUC = Area under the curve; CYP = Cytochrome P 450; P-gp = P-glycoprotein. White: No relevant DDI anticipated. Yellow: caution/careful monitoring required, especially in case of polypharmacy or in the presence of ≥2 yellow/bleeding risk factors. Orange: Consider dose reduction or avoiding concomitant use. Red: Contraindicated/not advisable. Blue dot indicates PK interaction.

**Table 12 pharmaceutics-14-01120-t012:** Predicted effects of anti-HIV therapies on DOACs exposure.

Concomitant Drug		Effect on DOACs Concentration			
Anti-HIV	Effect on P-gp and CYP	Dabigatran	Rivaroxaban	Apixaban	Edoxaban
HIV protease inhibitors	Strong CYP3A4 inhibition and P-gp inhibition or induction	● Variable increase and decrease in AUC	● +153% AUC +55% Cmax	● Strong increase in AUC predicted	● Strong increase in AUC predicted
DTG + ABC/TDF + 3TC	No relevant interactions known/assumed	No significant effect predicted			
DTG + TDF/TAF + FTC	No relevant interactions known/assumed	No significant effect predicted			
RAL + TDF/TAF + FTC	No relevant interactions known/assumed	No significant effect predicted			
EVGc + TAF/TDF + FTC	Cobicistat is a potent CYP3A4 and P-gp inhibitor	● Strong increase in AUC predicted			
DRVc + ABC + 3TC	Cobicistat is a potent CYP3A4 and P-gp inhibitor and darunavir is a CYP3A4 inhibitor	● Strong increase in AUC predicted			
DRVc + TDF/TAF + FTC	Cobicistat is a potent CYP3A4 and P-gp inhibitor and darunavir is a CYP3A4 inhibitor	● Strong increase in AUC predicted			
ATVc +TDF/TAF + FTC	Cobicistat is a potent CYP3A4 and P-gp inhibitor	● Strong increase in AUC predicted			
DRVr + TDF/TAF + FTC	Ritonavir is a potent CYP3A4 and P-gp inhibitor	● Strong increase in AUC predicted			
DRVr + ABC + 3TC	Ritonavir is a potent CYP3A4 and P-gp inhibitor	● Strong increase in AUC predicted			
EFV + TDF/TAF + FTC	Induction of CYP3A4 and P-gp	●Possible decreased exposure			
RPV + TDF/TAF + FTC	Induction of CYP3A4 and P-gp	●Possible decreased exposure			
AZT + 3TC + EFV	Induction of CYP3A4 and P-gp	●Possible decreased exposure			
TDF + 3TC/FTC + EFV	Induction of CYP3A4 and P-gp	●Possible decreased exposure			
TDF + 3TC/FTC + NVP	Induction of CYP3A4 and P-gp	●Possible decreased exposure			

3TC, lamivudine; ABC, abacavir; ATVc = atazanavir + cobicistat; AUC = Area under the curve; AZT, zidovudine; CYP = Cytochrome P 450; DRVc = darunavir + cobicistat; DRVr = darunair + ritonavir; DTG, dolutegravir; EFV, efavirenz; EVG, elvitegravir; FTC, emtricitabine; NVP, nevirapine; P-gp = P-glycoprotein; RAL, raltegravir; RPV, rilpivirin; TAF, tenofovir alafenamide; TDF, tenofovir disoproxil fumarate. White: No relevant DDI anticipated. Yellow: caution/careful monitoring required, especially in case of polypharmacy or in the presence of ≥2 yellow/bleeding risk factors. Red: Contraindicated/not advisable. Blue dot indicates PK interaction.

**Table 13 pharmaceutics-14-01120-t013:** Predicted effects of anti HCV therapies on DOACs exposure.

Concomitant Drug		Effect on DOACs Concentration			
**NS5A/B Polymerase Inhibitors**	**Effect on P-gp and CYP**	**Dabigatran**	**Rivaroxaban**	**Apixaban**	**Edoxaban**
Sofosbuvir	P-gp substrate	No significant effect on AUC predicted			
Ledipasvir	P-gp substrate and inhibitor	●Possible increase in AUC predicted			
Sofosbuvir + ledipasvir	P-gp/CYP3A4 substrate and moderate P-gp inhibition	●Possible increase in AUC predicted			
**NS5A/B-NS3/4A replication complex inhibitor**	**Effect on P-gp and CYP**	**Dabigatran**	**Rivaroxaban**	**Apixaban**	**Edoxaban**
Sofosbuvir + velpatasvir	P-gp/CYP3A4 substrate and moderate P-gp inhibition	●Possible increase in AUC predicted			
Sofosbuvir + velpatasvir + voxilaprevir	P-gp/CYP3A4 substrate and strong P-gp inhibition	● +160–180% AUC and C_max_	●Possible increase in AUC predicted	●Possible increase in AUC predicted	● Strong increase in AUC predicted
Ombitasvir + paritaprevir/ritonavir + dasabuvir	Ritonavir is a potent CYP3A4 and P-gp inhibitor	●Moderate increase in AUC predicted			
Elbasvir + grazoprevir	CYP3A4 and P-gp competition	●Possible increase in AUC predicted			
Glecaprevir + pibrentasvir	P-gp inhibition and competition	● +138% AUC +105% C_max_	●Possible increase in AUC predicted	●Possible increase in AUC predicted	●Possible increase in AUC predicted

AUC = Area under the curve; CYP = Cytochrome P 450; P-gp = P-glycoprotein. White: No relevant DDI anticipated. Yellow: caution/careful monitoring required, especially in case of polypharmacy or in the presence of ≥2 yellow/bleeding risk factors. Orange: Consider dose reduction or avoiding concomitant use. Red: Contraindicated/not advisable. Blue dot indicates PK interaction.

**Table 14 pharmaceutics-14-01120-t014:** Predicted effects of drugs used in the treatment of COVID-19 on DOACs exposure and pharmacological activity.

Concomitant Drug		Effect on DOACs Concentration and Pharmacodynamic Effect			
	Effect on P-gp and CYP	Dabigatran	Rivaroxaban	Apixaban	Edoxaban
Lopinavir + ritonavir	Strong CYP3A4 and P-gp inhibition	● Strong increase in AUC			
Darunavir + ritonavir or cobicistat	Strong CYP3A4 and P-gp inhibition	● Strong increase in AUC			
Atazanavir + ritonavir or cobicistat	Strong CYP3A4 and P-gp inhibition	● Strong increase in AUC			
Nirmatrelvir + ritonavir	Strong CYP3A4 and P-gp inhibition	● Strong increase in AUC			
Azithromycin	Mild P-gp inhibition	No PK data No dose reduction required			
Methylprednisolone and other corticosteroids	Moderate CYP3A4 induction; CYP3A4 competition	●No significant effect on AUC predicted Pharmacodynamically increased bleeding risk			
Tocilizumab	CYP3A4 and P-gp induction	●Possible decrease in AUC predicted			
Sotrovimab	No relevant interactions known/assumed	No significant effect on AUC predicted			
Regdanvimab	No relevant interactions known/assumed	No significant effect on AUC predicted			
Casirivimab + imdevimab	No relevant interactions known/assumed	No significant effect on AUC predicted			

AUC = Area under the curve; CYP = Cytochrome P 450; P-gp = P-glycoprotein. White: No relevant drug–drug interaction anticipated. Yellow: caution/careful monitoring required, especially in case of polypharmacy or in the presence of ≥2 yellow/bleeding risk factors. Orange: Consider dose reduction or avoiding concomitant use. Red: Contraindicated/not advisable. Blue dot indicates PK interaction and violet dot PD interaction.

## Data Availability

Not applicable.

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
