# Peer review of "Drug-Drug Interactions of Direct Oral Anticoagulants (DOACs): From Pharmacological to Clinical Practice"

_pharmaceutics, 2022, doi:10.3390/pharmaceutics14061120_

Round 1

Reviewer 1 Report

The article contains very important information. The article is very useful for doctors and scientists. Thanks to the authors.

Author Response

Comment: The article contains very important information. The article is very useful for doctors and scientists. Thanks to the authors.

Answer: We would like to thank the reviewer for the revision

Reviewer 2 Report

This review describes the potential drug-drug interactions of direct oral anticoagulants (DOACs) with several classes of drugs. The review is well written and the topic is of interest. However, I have some comments for the authors to address before considering this paper for publication.

The Authors should avoid writing “recommended”, “not advisable”, and similar phrases wherever the interactions are possible, not well-documented in the clinical trials conducted according to good clinical practice guidelines. This can be confusing to clinicians.

In reference to the previous one, the authors could clearly indicate the possible and clinically-relevant drug-drug interactions.

Abstract: "Their use has been demonstrated to be equally effective but safer than warfarin, -50% risk of hemorrhagic stroke, despite a significant increase of gastrointestinal bleeding" needs to be revised according to the whole picture.

The Authors could discuss the role of pharmacogenetics in drug-drug interactions of DOACs.

Author Response

Reviewer 2

Comment: The Authors should avoid writing “recommended”, “not advisable”, and similar phrases wherever the interactions are possible, not well-documented in the clinical trials conducted according to good clinical practice guidelines. This can be confusing to clinicians. In reference to the previous one, the authors could clearly indicate the possible and clinically-relevant drug-drug interactions.

Answer: We thank the reviewer for the comment. Clinical classification of the interactions was made after an appropriate evaluation of controlled clinical trials or, in absence of published data, on expert opinion derived from a scrupulous analysis of pharmacokinetic and pharmacodynamic characteristics of the drugs involved. Interactions are classified, as their clinical relevance, in 4 levels:

  1. White: No relevant DDI anticipated.
  2. Yellow: Caution required, especially in case of polypharmacy or in the presence of ≥ 2 yellow/bleeding risk factors.
  3. Orange: Consider dose reduction or avoiding concomitant use, careful monitoring required if combined.
  4. Red: Contraindicated/not advisable.

These levels are those used by the principal interaction databases and by the European Heart Rhythm Association Guidelines (Steffel J et al., 2021).

We have checked the manuscript and all the “recommended” and “not advisable” have been indicated on proved clinical interactions and indications.

Comment: Abstract: "Their use has been demonstrated to be equally effective but safer than warfarin, -50% risk of hemorrhagic stroke, despite a significant increase of gastrointestinal bleeding" needs to be revised according to the whole picture.

Answer: This sentence summarizes the conclusion of the meta-analysis from Ruff et al (Lancet 2014; 383: 955–62). We have now change it with “Their use has been demonstrated to have a favorable risk-benefit profile, with significant reductions in stroke, intracranial hemorrhage, and mortality compared to warfarin, but with increased gastrointestinal bleeding.”

Comment: The Authors could discuss the role of pharmacogenetics in drug-drug interactions of DOACs.

Answer: We agree with the reviewer that the pharmacogenetics of DDI with DOACs is a relevant topic. However, there are very few evidence reported in the literature of clear genetic predisposition to DDI with DOACs. There are also some reviews recently published on this topic (Kanuri SH et al., 2019; Lähteenmäki J et al., 2021; Liu Y et al., 2021; Raymond J et al., 2021; Shnayder NA et al., 2021; Tseng AS et al., 2018). We, therefore, decided to do not add an extra paragraph on our manuscript. We add the following sentence before the conclusions: An additional aspect that requires further investigation for a better prediction of DDI with DOACs is represented by the analysis of possible associations between genetic variants and PK profile. This topic was not covered in the present review but some important information can be found in a recent systematic review [133].

Reviewer 3 Report

Manuscript nicely captures the facts about DDI potential of various drugs when co-administered with DOAC’s. However, there are some flaws in the presentation and I would recommend making a few changes:

  • Please look for punctuation errors. Don’t break the word with a Hyphen and progress it to the next line. This disturbs the flow and also creates confusion.
  • Page 2, Line 61, Is it 5 to 17 hours. How is a half-life of 17 hours considered “short”? Also, line 64. Please correct it.
  • Introduction should be written more generalized, emphasizing the need for putting up this review. Please stay away from tables and also presenting too much data in the introduction section.
  • Please work on reducing the redundancy in the text.
  • Page 4, Line 126, correct “urines” to “urine”.
  • Is atenolol a Pgp susbtrate? It is largely considered a non-pgp substrate and low permeability marker. Addition of pgp inhibitor doesn’t alter its permeability suggesting that it is not a pgp substrate. Please clarify.
  • Color coding in tables is confusing. I would suggest using appropriate legends in the tables and footnoting them for clarity.
  • An expert discussion of the facts presented in the manuscript is missing. Please discuss the results from a point of each DOAC and the DDI potential with the multitude of drugs presented in the paper.
  • Conclusion section cannot be that big. Please concise it.

Author Response

Comment: Manuscript nicely captures the facts about DDI potential of various drugs when co-administered with DOAC’s. However, there are some flaws in the presentation, and I would recommend making a few changes:

  • Please look for punctuation errors. Don’t break the word with a Hyphen and progress it to the next line. This disturbs the flow and also creates confusion.

Answer: we have followed the layout of the journal which include the Hyphens. Punctuation errors were corrected, and a complete revision of the text was made.

  • Page 2, Line 61, Is it 5 to 17 hours. How is a half-life of 17 hours considered “short”? Also, line 64. Please correct it.

Answer: we have now cancelled “short” and indicated that the re-activation of the coagulation cascade occurs after approximately 24h after the interruption of the therapy.

  • Introduction should be written more generalized, emphasizing the need for putting up this review. Please stay away from tables and also presenting too much data in the introduction section.

Answer: We have now added the following sentence in the introduction “Considering their large use it is essential to have a complete picture on the possible drug-drug interaction (DDI) between DOACs and other commonly used classes of drugs. Although the last EHRA guidelines have nicely summarized the main DDI with DOACs [2], in the present review we have extend and updated the current knowledge on this topic that might be useful for clinician for their prescriptions.”

In addition, we divided in a different paragraph the pharmacokinetic and pharmacodynamic characteristics of DOACs.

  • Please work on reducing the redundancy in the text.

Answer: we have made substantial changes in the text and reduced the redundancies.

  • Page 4, Line 126, correct “urines” to “urine”.

Answer: We corrected it

  • Is atenolol a Pgp substrate? It is largely considered a non-pgp substrate and low permeability marker. Addition of pgp inhibitor doesn’t alter its permeability suggesting that it is not a pgp substrate. Please clarify.

Answer: we thank the reviewer for the comment. We have checked and the indication by Frost et al 2017 that atenolol was a substrate is indeed wrong. Atenolol is not a P-gp substrate. We therefore properly correct this statement.

  • Color coding in tables is confusing. I would suggest using appropriate legends in the tables and footnoting them for clarity.

Answer: we believe that color coding helps the readers to understand the level of DDI predicted.

White: No relevant DDI anticipated.

Yellow: Caution required, especially in case of polypharmacy or in the presence of ≥ 2 yellow/bleeding risk factors.

Orange: Consider dose reduction or avoiding concomitant use, careful monitoring required if combined.

Red: Contraindicated/not advisable

For clarity, in the footnote of each table there is a complete explanation of colors and acronyms used.

  • An expert discussion of the facts presented in the manuscript is missing. Please discuss the results from a point of each DOAC and the DDI potential with the multitude of drugs presented in the paper.

Answer: We agree with the reviewer. However, it is difficult to make an expert discussion for each DOACs. We decided to add a general expert opinion before the conclusions and to add a brief summary for each paragraph.

  • Conclusion section cannot be that big. Please concise it.

Answer: we concise the conclusions and utilized part of it for a final expert opinion statement.

Reviewer 4 Report

The authors present a review manuscript on the issue of DDIs between DOACs with other drugs. The thematic of this work is of great interest since the DOACs have found a widely usage as safer and more effective drugs replacing warfarin and analogues anticoagulant agents of choice for the long-term treatment of arterial and venous thromboembolic events or other thrombotic conditions. Thus this work would be of high interest for the readers. 

Unfortunately, my biggest consideration is that the manuscript shows high resemblance with the work of Steffel J, Collins R, Antz M, et al. 2021 European Heart Rhythm Association Practical Guide on the Use of Non-Vitamin K Antagonist Oral Anticoagulants in Patients with Atrial Fibrillation. Europace. 2021;23(10):1612-1676. doi:10.1093/europace/euab065. It is the reference [2] that is mentioned 17 times within the text and also the tables and text. So what is new and novel regarding this work? Is it an effort to summarize the information from reference 2? 

For example, the manuscript presents cases of DDIs without justifying enough why these drug categories while it does it more or less in the same way ref 2 is presenting them. The authors make a reference for comorbidities but it would be better if they described it further. Which are more prevalent in these cases, which drugs are often co-administered for patients administered DOACs etc. What is different from reference [2]. Apart of all this would assist the reader to keep in mind which drugs the review will focus on later. For example why antiarrhythmic drugs straight on after the aim of this work. It may be obvious but I believe a better presentation would benefit the manuscript. For example what level of evidence is there? What are the clinical guidelines etc. For example (considering their well written title) they could present the pharmacological perspective (mechanisms etc.) and then the clinical perspective (RCTs results guidelines etc.) in 2 sections. 
The way they decide to do it however substracts originality so I kindly ask them to justify and address this issue if a revised version is asked. In addition, we do not see what kind of information the authors were researched (level of evidence, type of studies etc.) or how the information were organized and presented in this work. The manuscript is simple presenting each category more or less as ref 2 does.  

Other comments: 

Please add the protein-binding in plasma lines 68-75 as additional factor for interactions. Especially for warfarin and the S-R enantiomer differences in protein binding it is a point of DDIs for anticoagulation. 

Table 1. Riaroxaban bioavailability is absolute or relative? 
Table 1. Better state halflife not half lifetime 

Lines 81-110 of the introduction. The authors present the PK characteristics for DOACs but also present DDIs (i.e. PPIs and food effects). I believe it may improve the reading process if edited so these two issues remain separately. 

Line 80 needs space in lines from main text. Same for line 110 and the other tables. 

There are grammatical errors throughout the text. Also there are paragraphs of one sentence (i.e., 95-97 and 135-136). Acronyms are not constant throughout the text (e.g., 510 pharmacokinetic, 513 pharmacodynamic)
That makes the manuscript difficult to read and maybe an edit in the text would greatly help its reading. 

2. Potential interactions with antiarrhythmic drugs. Digoxin and P-gp inhibitors and other cardiovascular drugs. Please edit. Place first the digoxin and other P-gp substrates of the same category and then discuss the inhibitors. It is confusing as it is because we return in digoxin later on for rivaroxaban. Line 165 states it clearly and should start with this and then explain why (mechanisms, studies etc.). The authors try to describe Table 3 but it is confusing the way they do it. Either describe each line for each DOAC or for each DOAC state the important information.

What are the blue dots on tables? The colors as well but I guess they are to show the important info on tables. I believe this could be better presented. 

Tables 1-12. Please provide relative references of the sources that the PKs or other DDI info were extracted.

Lines 310-322. I suggest make separate section of DOACs and herbal medicinal products or food supplements. 

Line 591. Please edit and state antiviral agents for human immunodeficiency and hepatitis C viruses

Author Response

Comment: Unfortunately, my biggest consideration is that the manuscript shows high resemblance with the work of Steffel J, Collins R, Antz M, et al. 2021 European Heart Rhythm Association Practical Guide on the Use of Non-Vitamin K Antagonist Oral Anticoagulants in Patients with Atrial Fibrillation. Europace. 2021;23(10):1612-1676. doi:10.1093/europace/euab065. It is the reference [2] that is mentioned 17 times within the text and also the tables and text. So what is new and novel regarding this work? Is it an effort to summarize the information from reference 2? 

For example, the manuscript presents cases of DDIs without justifying enough why these drug categories while it does it more or less in the same way ref 2 is presenting them. The authors make a reference for comorbidities but it would be better if they described it further. Which are more prevalent in these cases, which drugs are often co-administered for patients administered DOACs etc. What is different from reference [2]. Apart of all this would assist the reader to keep in mind which drugs the review will focus on later. For example why antiarrhythmic drugs straight on after the aim of this work. It may be obvious but I believe a better presentation would benefit the manuscript. For example what level of evidence is there? What are the clinical guidelines etc. For example (considering their well written title) they could present the pharmacological perspective (mechanisms etc.) and then the clinical perspective (RCTs results guidelines etc.) in 2 sections. 
The way they decide to do it however substracts originality so I kindly ask them to justify and address this issue if a revised version is asked. In addition, we do not see what kind of information the authors were researched (level of evidence, type of studies etc.) or how the information were organized and presented in this work. The manuscript is simple presenting each category more or less as ref 2 does.  

Answer: we thank the reviewer for the comment. We have now emphasized the novelty of our work in the introduction: “Considering their large use, it is essential to have a complete picture on the possible drug-drug interaction (DDI) between DOACs and other commonly used classes of drugs. Although the last EHRA guidelines have nicely summarized the main DDI with DOACs [2], in the present review we have extend and updated the current knowledge on this topic that might be useful for clinician for their prescriptions.”. Thus, we believe that our work represents a novel and useful document for readers and clinicians.

We also add a summary section for each paragraph that may help the readers to have a final statement of the DDI for each category of drugs. The levels of evidence have been described in detail in the text for each drug considered. A clear indication of it can be found in the manuscript. The information were presented and divided in terms of class of drugs and we collected all the available information on this topic for each class of drugs. We believe that our document is far more exhaustive that EHRA Guidelines. These European guidelines are frequently cited in the text because they are the principal source of information for a cardiologist, so, given that this work is addressed primarily to practitioner who usually prescribe DOACs, as cardiologists, it’s important to make a close comparison with data present in the guidelines. For the same reason, we choose to maintain a similar framework of the tables and similar colors to indicate the clinical relevance of the interactions. Despite these similarities, in this work there are many more data about interactions between DOACs and different classes of drugs compared to the work of Steffel J et al. Moreover, this work, in contrast to the guideline cited, contains a comprehensive discussion of the tables presented: each pharmacological class of interacting drugs is introduced and all the pharmacokinetic/pharmacodynamic and clinical data about each interaction are summarized.

We hope the reviewer will find the new version of the document improved and suitable for publication.

Other comments: Please add the protein-binding in plasma lines 68-75 as additional factor for interactions. Especially for warfarin and the S-R enantiomer differences in protein binding it is a point of DDIs for anticoagulation. 

Answer: Information about binding to plasmatic protein, already inserted in Table 1, were added also in the text. Protein binding are remarkably different between DOACs, with rivaroxaban and apixaban showing values above 90% and 87%, respectively. Thus, these two drugs may undergo to protein displacement by drugs with higher affinity to albumin and possible increase of their exposure. We recall that warfarin shows a very similar high protein binding (89%), and S(-) isomer has a slightly greater affinity than R(+)  [24].

Comment: Table 1. Riaroxaban bioavailability is absolute or relative? 

Answer: We thank the reviewer for the comment. The bioavailability in table is the absolute value.

Comment: Table 1. Better state halflife not half lifetime

Answer: we corrected it 

Comment: Lines 81-110 of the introduction. The authors present the PK characteristics for DOACs but also present DDIs (i.e. PPIs and food effects). I believe it may improve the reading process if edited so these two issues remain separately. 

Answer: We thank the reviewer. We move this sentence in the paragraph of DDI with antiacids.

Comment: Line 80 needs space in lines from main text. Same for line 110 and the other tables. 

Answer: We corrected it

Comment: There are grammatical errors throughout the text. Also there are paragraphs of one sentence (i.e., 95-97 and 135-136). Acronyms are not constant throughout the text (e.g., 510 pharmacokinetic, 513 pharmacodynamic)
That makes the manuscript difficult to read and maybe an edit in the text would greatly help its reading.

Answer: We have edited all text. We hope to have found all the acronyms and grammatical errors. 

Comment: 2. Potential interactions with antiarrhythmic drugs. Digoxin and P-gp inhibitors and other cardiovascular drugs. Please edit. Place first the digoxin and other P-gp substrates of the same category and then discuss the inhibitors. It is confusing as it is because we return in digoxin later on for rivaroxaban. Line 165 states it clearly and should start with this and then explain why (mechanisms, studies etc.). The authors try to describe Table 3 but it is confusing the way they do it. Either describe each line for each DOAC or for each DOAC state the important information.

Answer: we have re-write this paragraph. We hope that now is clearer and more logical.

Comment: What are the blue dots on tables? The colors as well but I guess they are to show the important info on tables. I believe this could be better presented. 

Answer: The meaning of the blue dots are described in the legend. “Blue dot indicates PK interaction and violet dot PD interaction”. We could not find any other way to indicate the type of interaction (PD and PK)

Comment: Tables 1-12. Please provide relative references of the sources that the PKs or other DDI info were extracted.

Answer: We believe that Tables 1-12 must be useful to a direct and simple identification of the clinical relevance of the interaction described. Adding references in the tables make them too busy and difficult to read. References of the various clinical trials presented are inserted in the text.

Comment: Lines 310-322. I suggest make separate section of DOACs and herbal medicinal products or food supplements. 

Answer: We intentionally excluded this topic since the clinical evidence are missing. There are few information available to make clear statements. Only St. John’s wort is cited for the relevance of the interaction with DOACs.

Comment: Line 591. Please edit and state antiviral agents for human immunodeficiency and hepatitis C viruses

Answer: We thank the reviewer for the comment. We have changed the title of the paragraph.

Reviewer 5 Report

This manuscript gave a comprehensive review of the drug-drug interactions between direct oral anticoagluants and potential comedications. Due to the potential severe side effects of DOACs and their broad use in the elderly patients this review is of great interest. Overall, this is an important and well written manuscript that I recommend to be accepted for publication after consideration of one revision as follows:

Although DDI has been observed or predicted between DOACs and many different drugs, these interactions can be summarized as either pharmacokinetic or pharmacodynamic interaction. The pharmacokinetic interactions are mainly due to interaction with P-gp and CYPs in the intestine and thus affecting bioavailability. It could be helpful to give a summary of these potential interactions in the Introduction section (as a table showing e.g. how a Pgp inhibitor or inducer could potentially affect the exposure of DOACs, etc.). This summary could be of interest for drugs not listed in the manuscript or newly developed drugs. The DDI risk could be evaluated based on their interaction with Pgp or CYP enzymes.

Author Response

Comment: Although DDI has been observed or predicted between DOACs and many different drugs, these interactions can be summarized as either pharmacokinetic or pharmacodynamic interaction. The pharmacokinetic interactions are mainly due to interaction with P-gp and CYPs in the intestine and thus affecting bioavailability. It could be helpful to give a summary of these potential interactions in the Introduction section (as a table showing e.g. how a Pgp inhibitor or inducer could potentially affect the exposure of DOACs, etc.). This summary could be of interest for drugs not listed in the manuscript or newly developed drugs. The DDI risk could be evaluated based on their interaction with Pgp or CYP enzymes.

Answer: We thank the reviewer for the comment. We have explained in the introduction the two types of possible drug-drug interaction as follow: “P-glycoprotein (P-gp) plays an important role in PK profile of all DOACs (Table 1) [25]. P-gp partially limits the disposition of these drugs by reducing their intestinal absorp-tion and facilitating their elimination by the kidney and the liver [26]. For this reason, potent P-gp inhibitors or inducers are expected to have relevant pharmacological interactions with all DOACs (Table 2), increasing or reducing their anticoagulant effect. Differently, different classes of drugs may have a pharmacodynamic (PD) interaction by affecting either the coagulation cascade or platelet activation. Thus, the DDI can be divided in PK and PD according to the different mechanism of interaction.”

“Both apixaban and rivaroxaban are partially metabolized in the liver by CYP3A4, therefore strong inducer and inhibitors of this hepatic cytochrome (Table 2) may influence the PK of the anticoagulants and alter their pharmacological effect.”

Round 2

Reviewer 3 Report

The comments were adequately addressed and the revised manuscript is good for publishing.

Author Response

We thank the reviewer for the final statement

Reviewer 4 Report

The authors presented an updated manuscript that tried to address the comments made from the first round of reviews. Prior to any more comments some things first. The authors should be congratulated for their long-detailed work. This is out of question. They discuss in their work a very important drug class that has gain the momentum in physicians’ preferred therapy standard of care in anticoagulation and it is very often that these patients have comorbidities and receive different medications. Thus, DDIs is an important part for optimum healthcare provision regarding COACs. This is something that the authors sufficiently point out in their work. I also understand their "novelty" answer, but I would expect a better representation so the "updated" knowledge to be more feasible in the way it is included and presented. Like, what new is gained through this work? The authors state " it’s important to make a close comparison with data present in the guidelines." Where within the text can we compare the updated information?

For example, a table or a decision tree (if this yes/no, then) would help to give some examples for co-administrations for cardiologists. Some clinical examples or debates over case reports and so on. That is why in my initial report I mentioned the high resemblance. Hence, regarding the closeness with the article in Europace. 2021;23(10):1612-1676. doi:10.1093/europace/euab065, it would be better if they could extract and summarize the information instead of following a similar format as within the guidelines that tend to be detailed and tiring to read sometimes.

Tables. The dot and color system are confusing and not useful so maybe should be replaced. The authors should revise them and present it in a better way. A simple “PK” or “PD” could be used. Arrows instead of increase decrease could assist (or +/- as they do in other cells), instead of no data could use N/A or a dash (-).

Under what terms the classification of clinically significance is presented in this work? What is the scientific difference when they state, "caution required" vs "careful monitoring"? Usually "use with caution/monitor" is used together to describe the significance. Also, what the authors mean by “significant interaction”? They could also use, major -moderate-minor or other ways to classify the DDIs. Is it simple transfer of how the interactions are reported in guidelines and other studies? What is the scientific opinion from the authors? They could have used a classified system in a uniform way throughout their text. There are guidelines (i.e., from FDA) of how to interpret the outcomes of PK studies (AUC fold increase or decrease etc.) but we do not see it here. The work would be interesting and of added value if they had presented in same standard way the outcomes form the PK studies especially since they want to provide an update of the available evidence.

Table 2 is a very small fraction of the actual inhibitors/inducers for CYP3A4 and P-gp. They use the same one from ref 30 for endoxaban (authors’ previous work) but probably could have created a new expanded one. See the Flockhart Table™ https://drug-interactions.medicine.iu.edu/MainTable.aspx see the FDA’s guidance etc.

Summaries. Why summaries and not one summary with the most important DDIs? That could add some value in the text as an updated version. Since they aim to provide a work that would be "useful for clinician for their prescriptions" I hope they would agree with me that in "summary" sections the terms "per os" or "oral" or "i.v." or parenteral administrations etc. could be mentioned. It is obvious that for DDIs we refer mostly for the systemic exposure but in a scientific work as this one it may need to be emphasized.

Why whole paragraphs with only one sentence? Lines 100-130 for example. Generally, English grammar and style needs improvements.

Line 135. What type of studies? In vitro? In vivo? Clinical? Line 140 what specific stands for? Is it only one study? Randomized or not? What level of evidence can we conclude we have? Also, a simple phrase would suffice. Digoxin has not showed any capability to modulate in clinically significant way the PK profiles for DOACs. As they do it later in lines 150-154.

These comments run through the whole manuscript. Although the authors with their work make a very good reference point their choices in the format, they followed makes it difficult to extract information in a feasible way.

At least, an update tables style for DDIs and a uniform style of significance of DDIs throughout the text would greatly benefit the manuscript. It would also show an added value with a review analysis of the available evidence, not only a simple report of existing knowledge. 

Overall, the authors should be congratulated for their long-detailed work. However, I believe that the manuscript has flows that subtract the importance of its content as well as its novelty and its general contribution in the field despite the very good work that the they made to gather all this information.

Author Response

Comment: The authors presented an updated manuscript that tried to address the comments made from the first round of reviews. Prior to any more comments some things first. The authors should be congratulated for their long-detailed work. This is out of question. They discuss in their work a very important drug class that has gain the momentum in physicians’ preferred therapy standard of care in anticoagulation and it is very often that these patients have comorbidities and receive different medications. Thus, DDIs is an important part for optimum healthcare provision regarding COACs. This is something that the authors sufficiently point out in their work. I also understand their "novelty" answer, but I would expect a better representation so the "updated" knowledge to be more feasible in the way it is included and presented. Like, what new is gained through this work? The authors state " it’s important to make a close comparison with data present in the guidelines." Where within the text can we compare the updated information?

Answer: We thank the reviewer for his comment. We apologize for the misleading answer, indeed, it was not our intention to make comparison between our document and EHRA guidelines. Our intention was to update and extend the info provided by the guidelines on DDI. 

Comment: For example, a table or a decision tree (if this yes/no, then) would help to give some examples for co-administrations for cardiologists. Some clinical examples or debates over case reports and so on. That is why in my initial report I mentioned the high resemblance. Hence, regarding the closeness with the article in Europace. 2021;23(10):1612-1676. doi:10.1093/europace/euab065, it would be better if they could extract and summarize the information instead of following a similar format as within the guidelines that tend to be detailed and tiring to read sometimes.

Answer: We thank the reviewer, but we believe that it is essential to start from the article in Europace and integrate the information. For the readers it is easier to have a document (our) with all the info on DDI instead of a simple integration document. Indeed, we extent but also correct some mistakes present in the guidelines. Regarding the decision tree, we believe that it is not useful for cardiologist. Instead, the classification of DDI in different colors are the easiest way to easily identify the possible or likely or certain DDI with DOACs.

Comment: Tables. The dot and color system are confusing and not useful so maybe should be replaced. The authors should revise them and present it in a better way. A simple “PK” or “PD” could be used. Arrows instead of increase decrease could assist (or +/- as they do in other cells), instead of no data could use N/A or a dash (-).

Answer: we disagree with this suggestion. We think that the system utilized in the guidelines are very clear and not confusing.

Comment: Under what terms the classification of clinically significance is presented in this work? What is the scientific difference when they state, "caution required" vs "careful monitoring"? Usually "use with caution/monitor" is used together to describe the significance.

Answer: We simply follow the indication provided by the EHRA guidelines. However, we now used the same indication as follow: “caution/careful monitoring required”

Comment: Also, what the authors mean by “significant interaction”? They could also use, major -moderate-minor or other ways to classify the DDIs. Is it simple transfer of how the interactions are reported in guidelines and other studies?

Answer: We provided, when available, the percentage of changes observed when two drugs are concomitantly administered. This info is very useful in order to understand the extent of the interaction. When we indicated ”significant”, we meant that the DDI can have a clinical impact. Major moderate and minor are indicated with colors: red, orange and yellow. We believe that this system is simple to read and easy to understand.

Comment: What is the scientific opinion from the authors? They could have used a classified system in a uniform way throughout their text. There are guidelines (i.e., from FDA) of how to interpret the outcomes of PK studies (AUC fold increase or decrease etc.) but we do not see it here. The work would be interesting and of added value if they had presented in same standard way the outcomes form the PK studies especially since they want to provide an update of the available evidence.

Answer: we actually provide this info by classifying the DDI in:

  1. White: No relevant DDI anticipated.
  2. Yellow: caution/careful monitoring required, especially in case of polypharmacy or in the presence of ≥ 2 yellow/bleeding risk factors.
  3. Orange: Consider dose reduction or avoiding concomitant use.
  4. Red: Contraindicated/not advisable.

 Again, this system has been used by EHRA and we believe that it must be used also in our manuscript for an easy comparison/extension of the info.

Comment: Table 2 is a very small fraction of the actual inhibitors/inducers for CYP3A4 and P-gp. They use the same one from ref 30 for endoxaban (authors’ previous work) but probably could have created a new expanded one. See the Flockhart Table™ https://drug-interactions.medicine.iu.edu/MainTable.aspx see the FDA’s guidance etc.

Answer: we are sorry but table 2 is different than those of edoxaban manuscript. This table indicate a class of P-gp inhibitors or inducers with different effect on CYP3A4. This info cannot be found in the Flockhart Table™ which indicates only P-gp OR CYP inhibitors.

Comment: Summaries. Why summaries and not one summary with the most important DDIs? That could add some value in the text as an updated version. Since they aim to provide a work that would be "useful for clinician for their prescriptions" I hope they would agree with me that in "summary" sections the terms "per os" or "oral" or "i.v." or parenteral administrations etc. could be mentioned. It is obvious that for DDIs we refer mostly for the systemic exposure but in a scientific work as this one it may need to be emphasized.

Answer: we add summaries by answering from previous revision. We believe that this is a simple way to get the most relevant info on DDI for each class of drugs. The administration is mainly oral. There are no sufficient examples to divide the DDI by type of administration.

Comment: Why whole paragraphs with only one sentence? Lines 100-130 for example. Generally, English grammar and style needs improvements.

Answer: we have extensively revised the manuscript and improved English

Comment: Line 135. What type of studies? In vitro? In vivo? Clinical? Line 140 what specific stands for? Is it only one study? Randomized or not? What level of evidence can we conclude we have? Also, a simple phrase would suffice. Digoxin has not showed any capability to modulate in clinically significant way the PK profiles for DOACs. As they do it later in lines 150-154.

Answer: we have cancelled this sentence. We have changed “specific” with clinical. We confirm that digoxin has not clinically relevant interaction with DOACs. Lines 150-154 are referred to beta blockers.

Comment: These comments run through the whole manuscript. Although the authors with their work make a very good reference point their choices in the format, they followed makes it difficult to extract information in a feasible way.

Answer: as indicated by the reviewer we have included all the available info on DDI with DOACs. We are sorry that the reviewer found difficult to extract information in a feasible way.

Comment: At least, an update tables style for DDIs and a uniform style of significance of DDIs throughout the text would greatly benefit the manuscript. It would also show an added value with a review analysis of the available evidence, not only a simple report of existing knowledge. 

Answer: We believe to have already provided a uniform style of significance of DDI in the text. This is simply indicated with different colors.  

Comment: Overall, the authors should be congratulated for their long-detailed work. However, I believe that the manuscript has flows that subtract the importance of its content as well as its novelty and its general contribution in the field despite the very good work that the they made to gather all this information.

Answer: we are sorry that the reviewer finds some flows. Our final goal was to summarize all the available info on DDIs and to provide an expert opinion on these interactions. We hope the reviewer would find the manuscript suitable for publication.